# Intrinsic Predictability Limits arising from Indian Ocean MJO Heating: Effects on tropical and extratropical teleconnections

David M. Straus[1], Daniela I.V. Domeisen[3,4], Sarah-Jane Lock[2], Franco Molteni[2], and Priyanka Yadav[5,6]

[1]Center for Ocean-Land-Atmosphere Studies, George Mason University, Fairfax, VA, USA
[2]European Centre for Medium-Range Forecasts, Reading, UK
[3]University of Lausanne, Lausanne, Switzerland
[4]Institute for Atmospheric and Climate Science, ETH Zurich, Zurich, Switzerland
[5]Global Modeling and Assimilation Office, NASA Goddard Space Flight Center, Greenbelt, MD, USA
[6]ESSIC, University of Maryland, College Park, College Park, MD, USA

**Correspondence:** David Straus (dstraus@gmu.edu)

**Abstract.** Since the Madden-Julian Oscillation (MJO) is a major source for tropical and extratropical variability on weekly to monthly timescales, the intrinsic predictability of its global teleconnections is of great interest. As the tropical diabatic heating associated with the MJO ultimately drives these teleconnections, the variability of heating among ensemble forecasts initialized from the same episode of the MJO will limit this predictability. In order to assess this limitation, a suite of 60-day ensemble reforecasts has been carried out with the ECMWF forecast model, spanning 13 starting dates from 01 November and 01 January for different years. The initial dates were chosen so that phases 2 and 3 of the MJO (with anomalous tropical heating in the Indian Ocean sector) were present in the observed initial conditions. The 51 members of an individual ensemble use identical initial conditions for the atmosphere and ocean. Stochastic perturbations to the tendencies produced by the atmospheric physics parameterizations are applied only over the Indian Ocean region ($50^o - 120^o E$). This guarantees that the spread between reforecasts within an ensemble is due to perturbations in heat sources only in the Indian Ocean sector. The point-wise spread in the intra-ensemble (or error) variance of vertically integrated tropical heating $Q$ is larger than the average ensemble mean signal even at early forecast times; however the planetary wave (PW) component of $Q$ (zonal waves 1-3) is predictable for 25 to 45 days, the time taken for the error variance to reach 50% to 70% of saturation. These scales never reach 90% of saturation during the forecasts. The upper level tropical PW divergence is even more predictable than $Q$ (40 to 50 days). In contrast, the PW component of the 200 hPa Rossby wave source, which is responsible for propagating the influence of tropical heating to the extratropics, is only predictable for 20 to 30 days. Substantial ensemble spread of 300 hPa meridional wind propagates from the tropics to the Northern Hemisphere storm-track regions by days 15-16. Following the growth of upper tropospheric spread in planetary wave heat flux, the stratosphere provides a feedback in enhancing the error via downward propagation towards the end of the reforecasts.

# 1 Introduction

The Madden-Julian Oscillation (Madden and Julian, 1971, 1972) is the dominant mode of tropical variability on subseasonal timescales of several weeks. The MJO manifests as a large-scale convection and precipitation pattern that starts in the Indian ocean, followed by an eastward propagation along the equator. Although the MJO itself is confined to the tropics, it can influence a large part of the globe, including the extratropics, through a wide range of remote impacts, so-called teleconnections (Stan et al., 2017). The best documented teleconnections include impacts on tropical cyclones (Maloney and Hartmann, 2000; Camargo et al., 2019; Hall et al., 2001; Camargo et al., 2009), the North Pacific (Wang et al., 2020), North America (Lin and Brunet, 2009), South America (Valadão et al., 2017), the North Atlantic region (Cassou, 2008; Lin et al., 2009), as well as the stratosphere over the Northern hemisphere polar regions (Garfinkel et al., 2014). The mechanisms for an MJO influence on global weather and climate act through the propagation of Rossby waves that are triggered by the diabatic heating anomalies associated with the MJO convection (Sardeshmukh and Hoskins, 1988).

MJO teleconnections are a major source of predictability on sub-seasonal to seasonal timescales around the globe (Vitart, 2017; Merryfield et al., 2020; Stan et al., 2022), including for tropical cyclones (Leroy and Wheeler, 2008; Lee et al., 2018; Domeisen et al., 2022), extreme precipitation (Jones et al., 2004; Muñoz et al., 2016), temperature in North America (Rodney et al., 2013), and the Northern Hemisphere stratosphere (Garfinkel and Schwartz, 2017), which in turn can have a strong downward impact on surface weather in the extratropics (Baldwin and Dunkerton, 2001), with strong impacts on extratropical sub-seasonal predictability (Domeisen et al., 2020a).

One class of well-studied tropospheric teleconnections relates to chances in the likelihood of occurrence of the North Atlantic Oscillation (NAO) . The phase of the MJO with enhanced convection over the Indian Ocean (MJO phases 2 and 3) is associated with the positive phase of the NAO (Lin et al., 2009; Ferranti et al., 1990; Cassou, 2008; Straus et al., 2015) roughly 1-2 weeks later, leading to added predictability (Lin et al., 2010; Vitart, 2014). The state of the NAO is also influenced by a stratospheric teleconnection pathway via upward planetary wave propagation (Garfinkel et al., 2014; Kang and Tziperman, 2018). Phases 2-3 (6-7) of the MJO suppress (enhances) the heat flux in the North Pacific region, resulting in a cooler (warmer) polar stratosphere and a stronger (weaker) polar vortex (Garfinkel et al., 2014; Schwartz and Garfinkel, 2020) The teleconnection of the MJO variability to the NAO is however also modulated by El Niño-Southern Oscillation (Lee et al., 2019), by the propagation speed of the MJO (Yadav and Straus, 2017), and by the strength of the Northern Hemisphere stratospheric polar vortex (Garfinkel and Schwartz, 2017).

Making extended range predictions for the mid-latitudes based on an MJO precursor remains challenging (Vitart et al., 2017; Schwartz and Garfinkel, 2020; Garfinkel et al., 2022; Stan et al., 2022), in part due to the large model errors associated with the teleconnections such as basic state bias (Schwartz and Garfinkel, 2020; Garfinkel et al., 2022; Stan et al., 2022; Schwartz et al., 2022). However, another limitation to predictability is the variability of heating among different observed episodes of a given phase, and the intermittency of heating and other sub-grid scale processes in space and time *even within a particular episode*. The dynamical character of the uncertainty in the response to this intermittency has been studied by Kosovelj et al.

(2019) in a low resolution (with T30 spectral resolution) global model using idealized stochastic parameterizations. Similarly, the response to model systematic error in Indian Ocean temperatures was addressed by Zhao et al. (2023).

The purpose of this paper is to extend the work of Kosovelj et al. (2019) to address the intrinsic limits of potential predictability due to the intermittency of heating and other sub-grid scale processes in a high resolution (50 km or less) operational forecast model setting. In this paper we do not address model systematic error. To this end a series of unique ensemble reforecasts have been carried out that are specifically designed to diagnose the uncertainty in both the tropics and the extratropics that arises due to the space/time intermittency of sub-grid scale processes within phases 2 and 3 of the MJO. In these reforecasts, made

with the Integrated Forecast System (IFS) of the European Centre for Medium-Range Forecasts (ECMWF), the stochastic parametrization scheme (SPPT) described in Leutbecher et al. (2017) has been altered so that perturbations that affect (directly or indirectly) diabatic heating tendencies are confined to the tropical Indian Ocean region. The SPPT alters the instantaneous tendencies of the temperature, specific humidity and horizontal wind components due to sub-grid scale physics processes by scaling these tendencies up or down in a stochastic manner. It is considered a standard component of the IFS. The physics pro-

cesses include those due to turbulent diffusion and sub-grid orography, convection, cloudiness and precipitation, and radiation. For more details, consult Leutbecher et al. (2017)

All members of each ensemble use the identical initial conditions, so that the only cause of model uncertainty (also called error here) must be the noise in heating introduced by the SPPT in the tropical Indian Ocean. We should point out here that one might design a similar set of experiments in which only the initial conditions were perturbed. (Such experiments could be used,

for example to understand the potential impact of changes in the observing system on error growth.) After all, any perturbation whatever to the system will quickly propagate and grow (Ancell et al., 2018). The question is how quickly such errors grow and saturate, and the answer depends on, among other factors, the amplitude of the initial errors. In fact, Zhang et al. (2019) use the this dependence of rate of growth on the magnitude of the initial condition error to estimate the predictability limit of mid-latitude weather. Judt (2020) study the dependence on the rate of initial condition error growth on region (tropical,

mid-latitude and high-latitude) for short simulations using a global storm-resolving model.

Another question regarding our experimental design is whether localizing the application of the SPPT to the tropical Indian Ocean is necessary, since following the argument of Ancell et al. (2018), perturbations in any region will quickly propagate to the Indian Ocean region. The only way to answer this question is to re-run the same set of experiments with SPPT applied globally, which is the subject of future research. We will return to this question in the Discussion section.

Since our goal is to document both the uncertainty in the tropical heating and the mid-latitude response in these experiments, we also consider the pathway by which the tropical heating forces extratropical Rossby waves. Although the MJO-related tropical heating is expected to force a corresponding signal in upper tropospheric divergence, this signal generally occurs within an easterly background wind, where stationary Rossby waves are not expected to propagate. Sardeshmukh and Hoskins (1988) derive a more complete formulation of the source of barotropic Rossby waves (hereafter Rossby wave source, or RWS)

which involves the advection of the absolute vorticity by the divergent component of the flow, and importantly acts in the vicinity of the sub-tropical jet, and so in a background of westerlies. Although strictly speaking the response to the MJO is not

stationary, previous success in explaining the extratropical response in terms of stationary wave theory (e.g. Matthews et al., 2004) suggests that stationary wave concepts are relevant here.

Section 2 describes the model and ensemble reforecast experiments in detail, as well as the methods used to diagnose the results. The signal and uncertainty in the tropical heating and Rossby wave source, as well as the growth of errors in the tropical circulation on various spatial scales, are described in Section 3. Section 4 describes the global spread of error in the circulation and its scale-dependence, while Section 5 shows results related to the stratospheric involvement. The Discussion is given in Section 6, and Conclusions in Section 7.

## 2 Methods and Data

This section introduces the data used in this study, the model experiments performed for this study, and the metrics used to evaluate the data.

### 2.1 Model configuration

This study has been performed using ensemble forecast experiments with the ECMWF Integrated Forecasting System (IFS Cy43r3: see ECMWF (2017a)). The IFS is a global earth system model, which includes an atmospheric weather prediction model coupled to ocean, sea-ice, ocean waves and land surface models. The experiments have been run with the ensemble fore-casting system ("ENS") close to the configuration used operationally for the ECMWF extended-range forecasts (https://www.ecmwf.int/en/forecasts/documentation-and-support/changes-ecmwf-model). This configuration has an atmospheric model res-olution of TCo319L91 (approximately 36 km horizontal grid spacing, with 91 vertical levels up to 0.01 hPa, with a timestep of 1200s), coupled to the NEMO ocean model v3.4.1 (Madec and the NEMO team, 2013) in the ORCA025_Z75 configuration (1/4 degree horizontal resolution, 75 vertical levels), the LIM2 sea-ice model (Goosse and Fichefet, 1999) and the ECMWF Wave Model (ecWAM ECMWF, 2017b). The model system used here is strongly related to the ECMWF extended-range pre-diction system used for subseasonal to seasonal (S2S) forecasts that is included in the S2S prediction project database (Vitart et al., 2017). This prediction system has been systematically evaluated for its ability to represent MJO teleconnections, along with other prediction systems from the S2S database (Vitart et al., 2017; Stan et al., 2022). Overall, the ECMWF system repro-duces a realistic teleconnection to the Northern Hemisphere, though with a too weak amplitude. Among the models investigated in the above studies, the ECMWF system exhibits a skillful representation of the MJO teleconnections.

### 2.2 Description of ensemble reforecasts

Ensemble reforecasts were made for initial dates of 01 November and 01 January, but only for those years (between 1981 and 2016) when the MJO resided in phases 2 or 3 with an amplitude greater than 1.0 on the initial date. From the MJO amplitude and phase data from the Australian Bureau of Meterology (http://www.bom.gov.au/climate/mjo/), this condition yielded eight years for the 01 November reforecasts and five for the 01 January reforecasts. The dates are given in Table 1. The large ensemble size used dictated that we keep the number of MJO-phase 3 initial dates to a relatively small number (here 13). In

order to make contact with previous reforecasts (the subject of a future publication), and to span varying parts of the seasonal cycle, 01 January and 01 November were chosen. However, we acknowledge that additional experiments would enable us to discriminate between the teleconnections in early and late winter, since they are known to be different, see e.g. Abid et al. (2021). All reforecasts initialized on 01 January (01 November) will be referred to as Jan (Nov) reforecasts.

All members of each ensemble forecast are initialized with *identical* initial conditions from the ERA-Interim reanalysis data (Dee and coauthors, 2011). There are no perturbations applied to the initial conditions. The single *control* forecast for each initial date also has no perturbations applied during the integration. The 50 additional members of each ensemble forecast differ only in that perturbations are applied during the model integrations. These are introduced via the operational model uncertainty scheme, SPPT (Stochastically Perturbed Parametrization Tendencies scheme, see Leutbecher et al., 2017; Buizza et al., 1999). SPPT is designed to represent model uncertainty due to the parameterization of atmospheric physical processes. In ECMWF's operational forecasts, SPPT perturbations are applied at every time step of the runs and at every grid-point over the entire globe. However, for these experiments, a mask is applied such that the SPPT perturbations are only active within a window over the Indian Ocean, i.e. they are fully active in $(50^oE - 120^oE, 20^oN - 20^oS)$ and tapered to zero within the neighbouring 5 degrees (in all directions). All the forecasts are run to a lead-time of 60 days.

In addition, Nov and Jan ensemble reforecasts were made for all years between 1981 and 2016 with the initial dates of 01 November and 01 January. Here the ensemble size is 9, with a control (unperturbed) run and 8 perturbed runs. Again the SPPT perturbations were confined to the tropical Indian Ocean. The purpose of these *all-year* reforecasts is to establish the model reforecast seasonal cycle for the November-December and January-February periods. Table 1 summarizes the MJO and all-year reforecasts, while Table 2 gives the MJO amplitude and phase for each of the reforecasts of Table 1. In addition, Table 2 gives the anomalies of the monthly mean Niño 3.4 indices to indicate the state of the El-Niño Southern Oscillation. This index is defined as the sea-surface temperature (SST) averaged over the region $5^o - 5^oN$ and $170^oW - 120^oW$, and was obtained from the website https://www.cpc.ncep.noaa.gov/data/indices/. Note that there are four 01 November start dates (for 1986, 1987, 2002 and 2015) which occur during a warm ENSO event, defined by having the Niño 3.4 index close to or above 1.0 for both forecast months. For the 01 January start dates, only 1987 and 2010 occurred during warm events. While the state of ENSO is known to affect the MJO teleconnections (Moon et al., 2011), we don't use a sufficient number of start dates to well sample the different ENSO phases separately. Thus the results we present represent averages over a variety of ENSO conditions.

## 2.3 Data and diagnostics

The output of the 60-day forecasts includes the fields of temperature $T$, geopotential height $Z$, horizontal winds $(u, v)$ and vertical pressure velocity $\omega$ at 12 pressure levels: $1000, 925, 850, 700, 600, 500, 400, 300, 250, 200, 100, 50\,hPa$. These fields were available twice-daily on an N80 Gaussian ($320^o \times 160^o$) lon x lat grid.

**Table 1.** Summary of the model runs performed for this study, for the November start dates (left) and the 01 January start dates (right).

| Start date | Ensemble size | Start date | Ensemble size |
|---|---|---|---|
| 01 Nov 1986 | 50+1 | 01 Jan 1987 | 50+1 |
| 01 Nov 1987 | 50+1 | 01 Jan 1990 | 50+1 |
| 01 Nov 1990 | 50+1 | 01 Jan 1995 | 50+1 |
| 01 Nov 2001 | 50+1 | 01 Jan 2010 | 50+1 |
| 01 Nov 2002 | 50+1 | 01 Jan 2013 | 50+1 |
| 01 Nov 2004 | 50+1 | | |
| 01 Nov 2011 | 50+1 | | |
| 01 Nov 2015 | 50+1 | | |
| 01 Nov 1981..2016 | 8+1 | 01 Jan 1981..2016 | 8+1 |

**Table 2.** Values of MJO amplitude and phase for the initial date of each reforecast ensemble, and observed monthly mean anomaly of the Niño 3.4 index for the two months of each reforecast ensemble.

| MJO Amplitude and Phase | | | | |
|---|---|---|---|---|
| Start Date | MJO Amp | MJO Phase | Nino3.4 Nov | Nino3.4 Dec |
| 01 Nov 1986 | 2.77 | 3 | 1.01 | 1.12 |
| 01 Nov 1987 | 1.58 | 2 | 1.07 | 0.94 |
| 01 Nov 1990 | 1.76 | 3 | 0.10 | 0.35 |
| 01 Nov 2001 | 1.56 | 3 | -0.37 | -0.41 |
| 01 Nov 2002 | 1.96 | 2 | 1.47 | 1.37 |
| 01 Nov 2004 | 1.32 | 3 | 0.66 | 0.74 |
| 01 Nov 2011 | 1.11 | 3 | -1.19 | -1.06 |
| 01 Nov 2015 | 2.04 | 3 | 2.72 | 2.66 |
| Start Date | MJO Amp | MJO Phase | Nino3.4 Jan | Nino3.4 Feb |
| 01 Jan 1987 | 1.22 | 2 | 1.14 | 1.13 |
| 01 Jan 1990 | 1.07 | 3 | 0.01 | 0.21 |
| 01 Jan 1995 | 1.41 | 3 | 1.02 | 0.73 |
| 01 Jan 2010 | 1.72 | 3 | 1.52 | 1.25 |
| 01. Jan 2013 | 1.07 | 3 | -0.53 | -0.52 |

The diabatic heating was computed as a residual in the thermodynamic equation, with resolution equivalent to T159 in spherical harmonic space, following the algorithm described in Swenson and Straus (2021). While the output of the algorithm yields the heating in $Wm^{-2}$ integrated over three layers: $1000-850hPa, 850-400hPa, 400-50hPa$, in this paper we present diagnostics from both the mid-level ($850-400hPa$) heating $Q_{mid}$ and the full vertical integral spanning $1000-50hPa$ heating $Q$. The daily mean diabatic heating from the ERA5 reanalysis (Hersbach and coauthors, 2020) was computed from the same fields, sampled four times per day, for the Novembers of the eight years listed in Table 1.

The Rossby wave source was computed following the prescription of Sardeshmukh and Hoskins (1988) as:

$$S = -\boldsymbol{\nabla} \cdot (\boldsymbol{v}_\chi \zeta_a) = -\zeta_a \boldsymbol{\nabla} \cdot \boldsymbol{v}_\chi - (\boldsymbol{v}_\chi \cdot \boldsymbol{\nabla}) \zeta_a \tag{1}$$

where $\zeta_a = f + \zeta$ is the absolute vorticity, and the vector $\boldsymbol{v}_\chi$ the divergent component of the horizontal flow vector. In words, $S$ is the sum of the stretching term (vorticity times divergence) and the advection of vorticity by the divergent flow. We shall attempt to diagnose the contribution of each term in Section 3 and the Appendix.

In order to compute $S$, we made use of the following transforms between the Gaussian grid and spherical harmonic (spectral) representations:

$$(u, v) \quad \rightarrow \quad (\hat{D}, \hat{V}) \tag{2}$$
$$(\hat{D}, \hat{V}) \quad \rightarrow \quad (u, v) \tag{3}$$
$$\hat{F} \quad \rightarrow \quad F \tag{4}$$

Here $(u, v)$ are the horizontal components of *any* vector on the grid, $(\hat{D}, \hat{V})$ are the corresponding divergence and curl in spectral representation, $F$ is the grid point representation of any scalar field and $\hat{F}$ its representation in spectral space. Applying transform 2 to the horizontal flow vector gives the ordinary divergence and vorticity in spectral space; transform 4 then yields the relative vorticity $\zeta$. To obtain the vector $\boldsymbol{v}_\chi$ we apply transform 3 using the spectral divergence $\hat{D}$ obtained from the horizontal winds but setting $\hat{V} = 0$. Finally we use 2 on the vector $(u_\chi \zeta_a, v_\chi \zeta_a)$ to obtain the corresponding divergence $\hat{S} = \hat{D}$ in spectral space, followed by at transform back to S. In applying this last transform, only spectral components corresponding to T21 were retained, and the final field was averaged over two-day blocks.

In order to consider the source outside the deep tropics (where the background easterlies would suppress a stationary wave response) and in the vicinity of the sub-tropical jet, we consider the average source between $15^o N$ and $30^o N$. The final values were divided by $\frac{2\Omega}{a}$, where $\Omega$ is the angular rotation rate of the earth, and $a$ the radius of the earth. The scaled values have units of $\frac{m}{s}$. The results shown in this paper are robust to changes in the latitude band chosen, both to modest poleward displacement and to widening it by 5 degrees.

### 2.4 Definition of anomalies

The seasonal cycles corresponding to the Nov and Jan MJO experiments listed in Table 1 are computed from the corresponding all-year experiments, and are characterized by a single climatological parabola in time for each variable and grid point, as in Straus (1983). Deviations in time about this seasonal cycle give the anomalies.

## 2.5 Metrics of uncertainty

Several metrics of the growth in uncertainty are used in this paper. In all cases what is measured is the uncertainty due solely to the spread of each ensemble with forecast time, without any reference to reanalysis or observations.

The *internal error variance* for a variable $F$ is defined as the average squared difference between the perturbed reforecasts (to which SPPT has been applied) and the control reforecast for the same initial date. It is denoted as $V_{int}^{(F)}$.

The *ensemble error variance* is defined as the average of the squared difference between the perturbed reforecasts and the ensemble mean. Its square root is referred to as the ensemble spread. It is denoted as $V_{ens}^{(F)}$.

Finally, the *external error variance*, denoted by $V_{ext}^{(F)}$ is defined as the average of the squared difference between each reforecast and all the control forecasts from the all-year experiments for the same forecast time.

All error variances depend on forecast time, as well as level, latitude and longitude (or zonal wavenumber). The external error variance gives a simple measure of the saturation level of the internal error variance. This saturation level will depend on time due to the evolution of the seasonal cycle.

To express these definitions in formulae, let the index $i$ denote ensemble member, running from $0$ (the control forecast) to $N = 50$, and let $F_{i,j}$ denotes the value of the field $F$ for forecast $i$ ($i = 0$ being the control forecast) for initial condition $j$. Let $\langle F \rangle_j$ denote the ensemble mean $\langle F \rangle_j = \frac{1}{N+1} \sum_i F_{i,j}$, and let $\tilde{F}_k$ denote the field $F$ from the control run for year $k$ of the all-year forecasts, with $1 \leq k \leq K = 36$.

$$V_{int}^{(F)} = \frac{1}{J} \sum_j \frac{1}{N} \sum_{i=1}^{N} (F_{i,j} - F_{0,j})^2 \tag{5}$$

$$V_{ens}^{F} = \frac{1}{J} \sum_j \frac{1}{N+1} \sum_{i=0}^{N} (F_{i,j} - \langle F \rangle_j)^2 \tag{6}$$

$$V_{ext}^{F} = \frac{1}{J} \sum_j \frac{1}{N+1} \sum_{i=0}^{N} \frac{1}{K} \sum_k \left( F_{i,j} - \tilde{F}_k \right)^2 \tag{7}$$

Note that the error variances of the kinetic energy are just the sum of the corresponding error variances of the components of the horizontal winds $(u, v)$, divided by 2.

## 3 Results

### 3.1 Tropical signal

The daily averaged evolution of vertically integrated diabatic heating anomaly is shown averaged for the 60-day experiments in Figure 1a. The heating has been averaged over the tropical band ($15^o S - 15^o N$), over all ensemble members and over all experiments. The eastward propagation of positive heating anomalies near longitude 90E can be seen for about 8 days, along with robust westward propagation of heating anomalies that appear after 4 days in the central Pacific. The ensemble spread of the heating (also averaged over the tropical band and all experiments) is shown in Figure 1b. The dominant influence of the

SPPT generated perturbations over the Indian Ocean sector is clear, leading to the largest ensemble spread in this sector. Note that the range of longitudes over which SPPT is applied is shown in the red vertical lines.

    Figure 2 shows the evolution of both the ensemble average of the latitudinally averaged RWS, averaged over all experiments, for the first 30 forecast days. Figure 2a shows the results for averaging over latitudes $15^o$N - $30^o$N, while 2b shows the results for latitudes $20^o$N - $35^o$N. In both cases, coherent eastward propagation is seen over the first 10 days with negative values seen

in at longitudes $100^o - 140^o E$. The magnitude of the RWS is larger for the band that extends to $35^o$N. Beyond forecast day 30, the mean signal in RWS shows little propagation (not shown).

    The colors indicate where the signal-to-noise ratio $\sigma$, defined as the ratio of the mean ensemble average to mean ensemble spread, is greater than 1.0 or (for positive RWS) or less than -1.0 (for negative RWS). This ratio shows that the propagation of the RWS is coherent (with the magnitude of the signal-to-noise ratio exceeding 1.0) for the first 8 to 10 forecast days.

Figure A2 shows the evolution of the two components of the RWS, the stretching term and the advection of vorticity by the divergent flow (see equation 1). While the stretching term generally dominates, the eastward propagating advection term to the east of the heating maximum is seen to contribute most to the signal at early forecast times.

### 3.2   Tropical error growth

The daily averaged ensemble spread of vertically integrated heating $Q$, averaged over all experiments, is shown in Figure 1b.

During the first few days, the heating spread grows substantially between longitudes $90^o$ and $95^o$ in the Indian Ocean, precisely where the mean heating is greatest. The values of the maximum spread decrease after about 6 days, but large spread is seen over a wider area as the uncertainty propagates. (Note for reference that the longitudinal boundaries of the SPPT perturbations are indicated by the faint vertical red lines in Figure 1b.)

    In order to determine whether the strength of the stochastic perturbations (reflected in the magnitude of the ensemble spread

in the Indian Ocean region) is reasonable, we also computed the inter-annual standard deviation $\sigma_{IA}$ of the tropical $Q$ from the ERA5 reanalysis over the eight years corresponding to the Nov experiment for the first 30 days. Figure A1 in the Appendix shows the daily evolution of the standard deviation with the same scale as in Figure 1b. The ERA5 $\sigma_{IA}$ is largely confined to the same regions as the model spread: the Indian Ocean region and the west-central Pacific. In the Indian Ocean sector, the model spread in heating has somewhat lower maximum values than $\sigma_{IA}$, but extends over a wider area. The model spread is

notably less than $\sigma_{IA}$ over the Pacific up to forecast day 30.

    In order to investigate the scale-dependence of the uncertainty of the heating evolution, we calculated the zonal wavenumber spectra of the internal error variance of $Q_{mid}$ ($850 - 400\,hPa$) averaged over 2-day blocks and over the tropical belt $15^o S - 15^o N$. Figure 3 shows the spectra for the blocks ending on days 2, 4, 6, 10, 20, 40 and 60. The latter is indicated by the red line, and can be compared to the external error (a measure of saturation) in the blue line. By day 2, the spectrum is relatively

flat down to length scales of about 3000 km, consistent with the variance being forced by perturbations in a narrow longitude range. As time progresses, the variance increases without dramatic change in shape until after day 20, when the larger scales (wavelengths greater than about 3500 km) grow more rapidly than the smaller scales, leading to a steeper variance spectrum by

60 days. If we define the predictability time $\tau$ to be the time it takes for the error variance to reach a given fraction of saturation, $\tau$ increases with length scale.

To make this more precise, we show the time $\tau$ at which the error variance of $Q$ reaches a fraction $f_\tau$ of the variance of the external error for $f_\tau = 0.50$, 0.70, and 0.90, as a function of zonal wavenumber in Figure 4a. The red curves give the results for $Q$ in the solid, dashed and dotted lines, respectively. Prior to calculating the error variances for this plot, we have truncated $Q$ (and the other fields to be shown) to a spherical harmonic T21 representation in order to eliminate excessive noise. $\tau$ increases with zonal scale (decreasing wavenumber) for all choices of $f_\tau$, but this is particularly noticeable for $f_\tau = 0.70$ and 0.90. In

fact the limit of 0.90 of the external error is never reached for zonal wavenumbers 1 and 2.

    The predictability times for the T21 representation of the upper-level (200 hPa) divergence are shown in Figure 4a in the blue curves. These times are notably longer than for the vertically integrated heating. For example, the divergence $\tau$ corresponding to $f_\tau = 0.70$ is greater than 35 days for the largest scales, compared to 20 days for the heating $\tau$. While this might be taken to indicate high predictability for the extra-tropical response, the corresponding times for the Rossby wave source, shown

in the blue curves in Figure 4b, are considerably shorter than those for $Q$. This is especially true for the planetary waves (wavenumbers 1 - 3) for which the predictability time for the RWS is shorter than that for the heating by about 8 days for $f_\tau = 0.50$, and by about 10 - 20 days for $f_\tau = 0.70$. This reflects the sensitivity of the RWS to the sub-tropical divergent flow and also the sub-tropical absolute vorticity (as expressed in equation 1). The RWS is the sum of the stretching and advection components. Hence the error measures, which are quadratic, will have contributions not only from each component but also

from their interaction. Figure A3 shows the total $V_{ens}^{(RWS)}$ as a function of zonal wavenumber and time. The interaction term (labeled NonLin in the figure) was computed by taking the difference between the total error and the stretching and advection components. For zonal wavenumbers greater than about 5, the error in the stretching term dominates, but for larger scales (lower wavenumbers) there is considerable compensation between the stretching and advection terms. The analysis of predictability times for the RWS has not, to our knowledge, been shown before, and is an important result regarding the predictability of the

extra-tropical circulation.

### 3.3  Global error growth

The forecast evolution of the spectra of the internal error variance of kinetic energy (KE) is presented in Figure 5 for different latitude bands. In the tropics ($15^o S - 15^o N$; Figure 5d), the spectra grow without much change in shape between forecast days 3 and 10. Between days 10 and 20 the spectra for smaller scales approach saturation much more quickly than do the larger

scales. Wavelengths shorter than about 3500 km are already saturated by day 20, while the larger scale error variance continues to grow beyond day 40. The times for which the spectra are shown in Fig. 5d (3, 5, 10, 20, 40 and 60 days) are the same as in the other panels (Figs. 5a - 5c), which show latitude bands of $25^o N - 35^o N$, $45^o N - 55^o N$ and $65^o N - 75^o N$, respectively. As one goes to higher latitudes (i.e. from panels (c) to (b) to (a)), the error variance curve for days 3 and 5 continuously moves to lower values, indicating the time it takes for the error to propagate poleward from the tropics. However by day 40, the curves

are at about the same level for all latitudes except the $65^o N - 75^o N$ band, where the error is less than at other latitudes. The saturation error (estimated by the red curve at day 60) is also less.

To get a sense of how the errors spread geographically, we present maps of the ensemble spread of the meridional wind in Fig. 6. The choice of meridional wind was motivated by its close relationship with storm tracks and circumpolar wave guides (Branstator and Teng, 2017). Much of the tropics outside of the Indian Ocean region is nearly error free even at day 6. By day 10, substantial error has already appeared in the extra-tropics, particularly in the storm-track regions, and by day 16 the extra-tropical spread has almost saturated. By day 30 the spread in the extra-tropics has essentially reached its saturation value, since it doesn't increase for longer lead times (not shown).

## 3.4   Error propagation through the stratosphere

Since the stratosphere plays a role in the teleconnections from the tropics as a modulator for extratropical long-range forecasts (e.g. Domeisen et al., 2015), we evaluate the role of the stratosphere in error propagation. Since the polar vortex tends to form in mid- to late-winter (Balwin and Holton, 1988) we present results for the Nov and Jan reforecasts separately. The ensemble error variance of the zonal wind $u$ was integrated over the polar cap ($60^o N - 90^o N$), and the contributions to the zonal mean error variance from errors in the zonal flow and zonal wavenumbers $m = 1 - 3$ computed. The corresponding spread (square root of the variance) is shown in Figures 7a and 7b for the Nov reforecasts and Figures 7c and 7d for Jan reforecasts, at levels from $1000 - 50 hPa$. Some evidence for the downward propagation of errors (in the form of ensemble spread increasing from the top, i.e. from $50 hPa$) from the stratosphere appears during the last 10 days of the November experiments, and slightly earlier in the January initializations, in both the zonal mean and planetary wave contributions.

The upward propagation of the wave activity from the upper troposphere into the stratosphere, as measured by the vertical component of the Eliassen-Palm flux, is proportional to the zonal mean of the meridional eddy heat flux. Thus we would expect that if the spread in the tropospheric planetary wave activity is responsible for the growing ensemble error in the stratosphere, we should see evidence for this in the spread in the heat flux, especially in the contribution from zonal wavenumbers $m = 1 - 3$. Figure 8 shows this heat flux as a function of time and pressure level for both Nov and Jan experiments. For the Nov experiments, enhanced spread in the eddy heat flux is seen by day 30, and by day 40 this spread has grown in the stratosphere. This occurs even earlier (by 10 days) in the Jan experiments, likely due to the stronger wave flux into the stratosphere and hence a stronger upward coupling. For the Nov experiments (Figure 8a), the spread increase in the upper troposphere is seen slightly prior to its increase in the stratosphere, although for the Jan experiments the increased spread in the troposphere and stratosphere tends to occur nearly simultaneously.

In order to better understand the longitudinal dependence of the upward error growth, the geographical distribution of the planetary wave heat flux at 50 hPa is described in Figure 9. This figure shows both the ensemble mean eddy heat flux due to $m = 1 - 3$ and the ensemble spread, whose zonal mean is depicted in Figures 8a. and 8b. The four rows give pentad time averages for pentads 1, 3, 5 and 7. The heat flux itself is largely confined to the North Pacific in the ensemble mean, which is the region of upward propagation of planetary waves from troposphere into the stratosphere in the MJO teleconnections (Schwartz and Garfinkel, 2020) but other high latitude regions contribute substantially to the spread. This is true particularly for the Jan experiments (columns 3 and 4), in which large values of the spread are seen over the entire belt around $60^o N$ by

310 pentad 5 (forecast days 21-25). The analysis of geopotential height spread at 500hPa in the North Pacific sector (not shown) gives similar results.

## 4 Discussion

The evolution of tropical heating for El-Niño years (not shown) indicates less eastward propagation from the Indian Ocean compared to normal years, in line with the findings of Liu et al. (2020), likely because less moisture is available over the
315 Indian Ocean due to the ENSO convection in the central Pacific. Nevertheless, the average over all forecasts does show distinct eastward propagation for the first 10 days or so. The Rossy wave source shown in Figure 2 shows a corresponding propagating signal over the broad Indian Ocean region for the first 10 days.

The evolution of the average of the ensemble spread in vertically integrated heating ($\Delta Q$) shown in Figure 1d shows clearly that the within-ensemble variability induced by the application of the regionally confined SPPT remains mostly confined to
320 that region ( $50^o - 120^o$E) for the first 10 days or so. This is also true for the tropical meridional wind spread (Figure 6) for the first 6 days. By day 10 the perturbations in mid-latitudes cover all longitudes. The early confinement of the errors to the tropics suggests that the evolution of the tropical heating and circulation uncertainties would be different had the SPPT been applied throughout the tropical belt. Whether this difference would strongly affect the growth of uncertainty in the extra-tropics is hard to assess directly from these experiments.

Consistent with the tropical $\Delta Q$ remaining somewhat regionally confined, its spectrum is relatively flat over a range of scales for the first 20 days or so (Figure 3), with the error growth similar for different scales. However, after day 20 scales less than a few thousand km. approach saturation, while the largest scales remain far from saturation. Here the saturation spectrum (given by the blue line) has become steeper, with the largest scales having the greatest variance.

The corresponding times predictability times $\tau$, shown in Figure 4, reveal an interesting result: The upper-level tropical
divergence, largely forced the tropical heating, is far more predictable than the heating, with the largest scales reaching 0.50 (0.70) of their saturation level only near day 40 (50). Clearly the upper-level divergence is relatively insensitive to the details of the heating. A naive interpretation of the tropical divergence as the main forcing function for the extra-tropics would indicate long-range predictability related to the MJO. However, the predictability times for the Rossby wave source $S$ are considerably shorter: the largest scales reach 0.50 (0.70) of saturation already at around 20 (30) days. This is understandable since $S$ is
influenced not only by tropical and subtropical divergence but also by the meridional gradient of the jet. The dominance of the stretching term for moderate scales shown in Figure A3 is consistent with strong upper-level divergence associated with baroclinic disturbances near the Pacific jet, as expressed by an increase in frequency of warm conveyor belt outflows (Quinting et al., 2023), and also by the behavior of the longitude-time plots of the stretching term that show very consistent eastward propagation (not shown).

Thus we can conclude that one path by which mid-latitude and sub-tropical variability may affect the response to tropical forcing is by changing the effective source for that response. Nevertheless, predictability times for $S$ of 20 - 30 days are

long enough to justify the approach of, for example, Matthews et al. (2004) in using stationary wave concepts to describe the extratropical response to the MJO.

The slopes of the saturation (or background) kinetic energy spectra presented in Figure 5 are compared to those corresponding to a dimensional wavenumber dependence of $k^{-3}$ and to a dependence of $k^{-5/3}$ as indicated by dashed and solid lines in the Figure. (The dimensional wavenumber $k = \frac{2\pi}{\lambda}$ with $\lambda$ being wavelength.) A slope corresponding to a $k^{-3}$ dependence is evidence of the dominant of rotational flow, while a $k^{-5/3}$ dependence in associated with the dominance of convection and gravity waves, and in general divergent flow (Charney, 1971; Sun et al., 2017; Zagar et al., 2017; Li et al., 2023).

Our tropical saturation spectrum, showing a $k^{-3}$ dependence, is in distinct contrast to that of Judt (2020) (their Figure 5), which shows a slope roughly corresponding to a $k^{-5/3}$ dependence, indicative of the dominance of convection and divergent flow. In mid-latitudes, the background spectrum of Judt (2020) also shows a transition from a $k^{-3}$ to a $k^{-5/3}$ dependence at several hundred km., a transition our results are unable to resolve. These differences are due to the model resolution and dynamics, as J uses 4-km horizontal resolution, storm-resolving simulations without convective parameterizations, while the IFS model used here has a resolution of 36-km and makes use of parameterizations for unresolved processes. Similar to the spectra shown in Zhang et al. (2019) (their Figure 6), the mid-latitude error growth is rapid between days 5 and 10, while the continued error growth for the largest scales at later times is similar to that reported in Selz (2019) in their Figure 4, noting that the spectra are plotted differently.

The growth of uncertainty in the stratospheric circulation, as seen in Figure 7, is forced by the upward propagation of the planetary wave meridional flux of sensible heat (which is the dominant term in the vertical component of the Eliassen-Palm flux), shown in Figure 8. This uncertainty then propagates downward into the upper and middle troposphere. While most of the upper troposphere sensible heat flux is due to planetary wave disturbances in the Pacific, its uncertainty in the North Atlantic and Asian sectors are also large, especially for the Jan experiments (Figure 9). This downward propagation is potentially linked to wave-mean flow interaction which acts to bring anomalies in e.g. wind and temperature to the lower stratosphere. The planetary wave error in the upper troposphere ($300hPa$) for Nov. reaches a maximum 20 days earlier than does the error at $50hPa$, hinting at a tropospheric forcing of the stratospheric spread.

The stratospheric descent of error seen in Figure7 occurs towards the end of the experiments, consistent with the tropospherically forced uncertainty being modulated by the stratospheric circulation (Domeisen et al., 2020b). This descent is seen about 10 days later in the reforecast period for the Nov experiments than for the Jan experiments. One factor contributing to this difference is the seasonality of the anomalies of height in the Pacific during Phase 3 of the MJO (Wang et al., 2023) which leads to a greater upward component of Eliassen-Palm flux seen for the Jan experiments in Figure 9. Another factor is likely to be the lack of a fully formed stratospheric vortex during November, so that the establishment of a wave guide for vertically propagating Rossby waves is delayed. (It was not possible to verify this since data were retained only up to $50hPa$.) Note however that separating the effects of the MJO from that of ENSO and other conditions such as stratospheric sudden warmings is quite difficult (Domeisen et al., 2019).

## 5 Conclusions

The suite of ensemble reforecast experiments presented here was explicitly designed to gauge the effect of the intrinsic uncertainty of sub-grid motions on the response to the MJO in phases 2 and 3. Each ensemble has all its members initialized identically during an observed MJO event, and differ from each other only in the realization of the stochastic parameterizations, applied only in the tropical Indo-Pacific region. Thus even though the errors (deviations within the ensemble) spread globally, they are ultimately due to the uncertainty in this region. These subsequent errors in the tropical diabatic heating, tropical upper-level divergence and Rossby wave source indicate the path towards mid-latitude uncertainty in the circulation response. We should point out that our results for the evolution of the Rossby wave source and its error growth hold only for MJO phases 2-3. Since the MJO modulates the Pacific storm track (Lee and Lim, 2012), the behavior of the RWS will be different in other phases of the MJO.

Caveats to this study include the dependence on the particular model used (the IFS) and the lack of comparison to experiments in which the perturbations were applied globally, as in the operational extended-range predictions of ECMWF, or in other regions. The IFS parameterizes convection and other sub-grid scale processes, possibly accounting for the difference between the tropical error spectra seen in these experiments (Figure 5) and those seen in a convection-resolving model (Judt, 2020). Although we have evidence that the perturbations initially confined to the tropical Indo-Pacific region takes at least 10 days to propagate to other tropical regions (Figure 1), we have not conducted parallel experiments in which the error was applied over a broader tropical band. These await the future.

In conclusion we find that:

- The uncertainty (average ensemble spread) in total vertically integrated heating is roughly the same magnitude as the MJO signal for the first 5 days, but overtakes the signal subsequently. [Figure 1].

- The propagation of the full Rossy wave source is quite coherent in space out to 10 days.[Figure 2]

- The uncertainty of tropical heating is highly scale-dependent: the error on planetary scales has not fully saturated even at the end of the 60-day forecasts, while the predictability time corresponding to 50% (70%) of saturation in about 25 (45) days. For scales of roughly $1500km$ (zonal wavenumbers 18-20), those times are reduced to 15 to 20 days. [Figures 3 and 4].

- While the predictability times for the planetary wave upper-level divergence are longer than those for diabatic heating (40 to 50 days), the full Rossby wave source is less predictable than the heating (planetary wave predictability times of 20 to 30 days). [Figure 4]. However this is long enough to validate the application of stationary-wave theory to the extratropical response, as in Matthews et al. (2004). The analysis of the predictability of the Rossby wave source is new.

- The kinetic energy error spectra show the spread of error from the tropics to the sub-tropics, mid-latitudes and high latitude, with the error at a give time decreasing as latitude increases [Figure 5].

- The ensemble spread in meridional wind generated over the tropical Indian Ocean amplifies and propagates into the extratropics, reaching a noticeable amplitude in the mid-latitude storm tracks after approximately 15 days, after which its amplifies in situ during the following week [Figure 6].

- The role of the stratosphere in amplifying uncertainty is generally confined to the latter part of the 60-day reforecasts, after the ensemble spread in upper-tropospheric heat flux has affected levels above 50 hPa [Figures 7 and 8].

*Code and data availability.* The computer codes used to create the Figures are written in Fortran, and were compiled a recent version of the Intel compiler. They are available from DMS by request. The data from ERA5 reanalysis are from the Computational and Information Systems Laboratory (CISL) of the National Center for Atmospheric Research. Users are required to obtain a Data Analysis Allocation. More information is at : https://arc.ucar.edu/xras_submit/opportunities. There were a total of 6 different IFS experiments to generate all the model data that are used in the paper. They are available on the following web pages:

- https://doi.org/10.21957/ms6x-gk09

- https://doi.org/10.21957/qtqh-5r32

- https://doi.org/10.21957/tzgp-tv45

- https://doi.org/10.21957/cf3y-0343

- https://doi.org/10.21957/ndqr-vs12

- https://doi.org/10.21957/kt7k-1r77

The DOIs link to a webpage that provides a description of the available data and retrieval scripts to access the data.

*Author contributions.* The experiments were designed by FM and carried out by SJL, both of whom oversaw the data storage. DMS carried out the analysis to create Figs. 1 through 9, while PY carried out the analysis for Figs. 10 through 12. DD wrote much of the Introduction, while DMS, SJL and PY contributed to the text of the paper. All authors contributed to the discussion of the results and feedback on the manuscript.

*Competing interests.* At least one of the (co-)authors is a member of the editorial board of Weather and Climate Dynamics.

*Acknowledgements.* Support from the Swiss National Science Foundation through project PP00P2_198896 to P.Y. and D.D. is gratefully acknowledged. The authors also acknowledge suggestions from Kai Huang, and helpful comments from Peter Dueben and Magdalena Alonso Balmaseda, as well as the anonymous reviewers. In addition, Barry Klinger provided valuable help in plotting the data for the spectra of kinetic energy and heating.

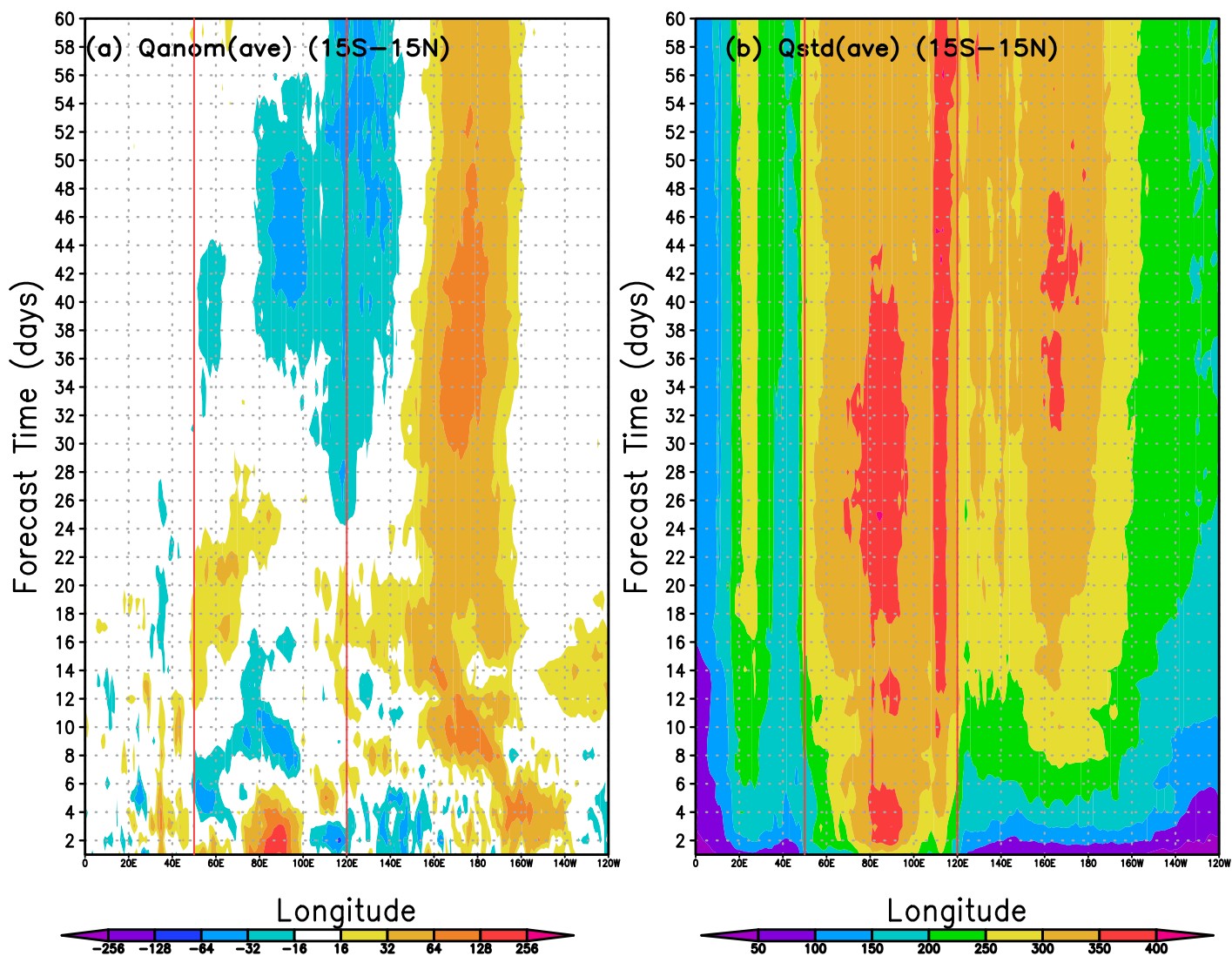

**Figure 1.** (a). Evolution of the daily mean, ensemble mean anomaly of diabatic heating anomaly $Q$ (averaged $15^o$ S-$15^o$ N) for days 1-60 of the 60-day experiments, averaged over all experiments. (b). The evolution of the ensemble standard deviation of the daily mean heating (vertically integrated and averaged $15^o$S-$15^o$N) averaged over all experiments. The abscissa gives the forecast time in days. The red lines indicate the range of longitudes over which the stochastic parametrization was applied. Units of $Wm^{-2}$.

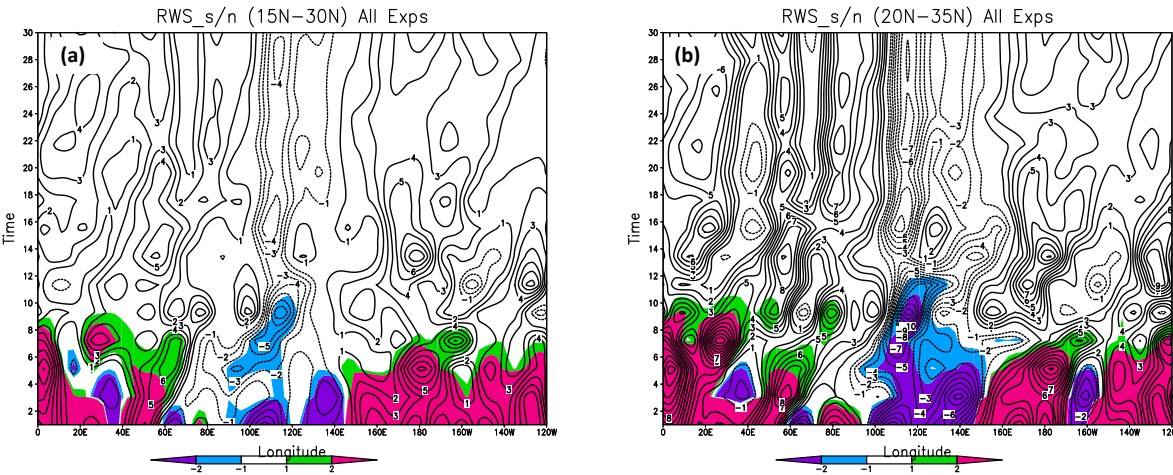

**Figure 2.** Evolution of the Rossby Wave Source ($S$) averaged over all experiments. Time is given in days. The RWS was computed at the equivalent of T21 triangular spectral truncation (see text for details). $S$ was averaged between $15^oN$ and $30^oN$ (panel a) and between $20^oN$ and $35^oN$ (panel b). The color scale gives the ratio of the ensemble mean to ensemble spread. The units of the RWS are $ms^{-1}$.

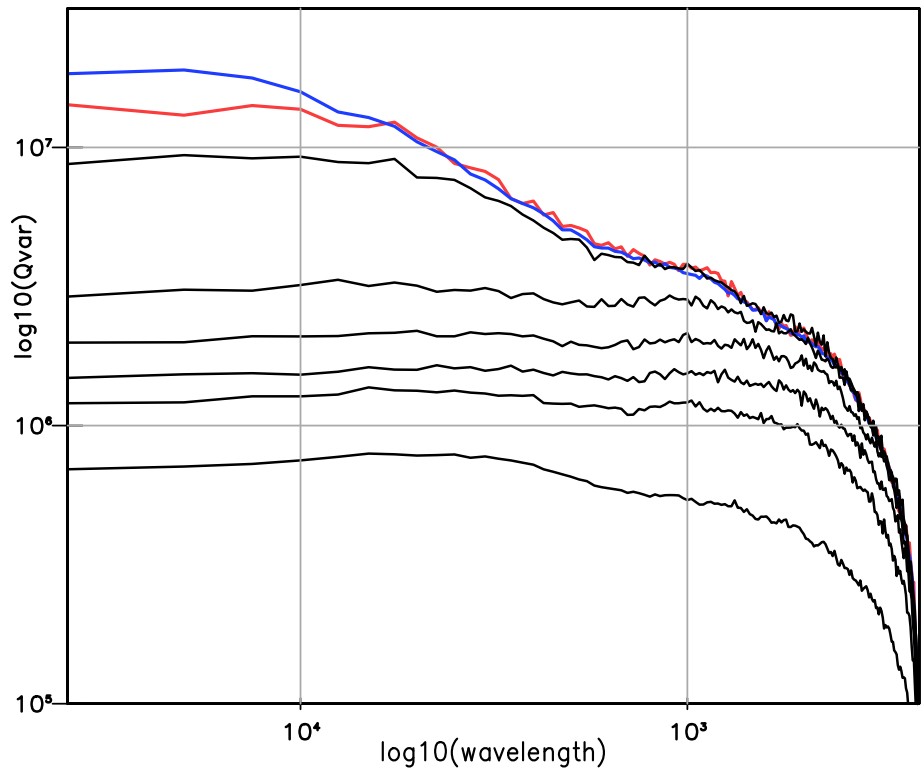

**Figure 3.** Zonal wavenumber spectra of internal error in mid-level tropical diabatic heating $Q_{mid}$ for various forecast times. The heating was averaged in 2-day blocks, and was averaged over latitudes $15^o$ S - $15^o$ N. The six black lines give the spectra (starting at the lowest line) for days 1-2, 3-4, 5-6, 9-10, 18-20, 39-40. The heating for days 59-60 is shown in the red line, and the saturation (external) error in the blue line. Units of $log_{10}(Wm^{-2})$.

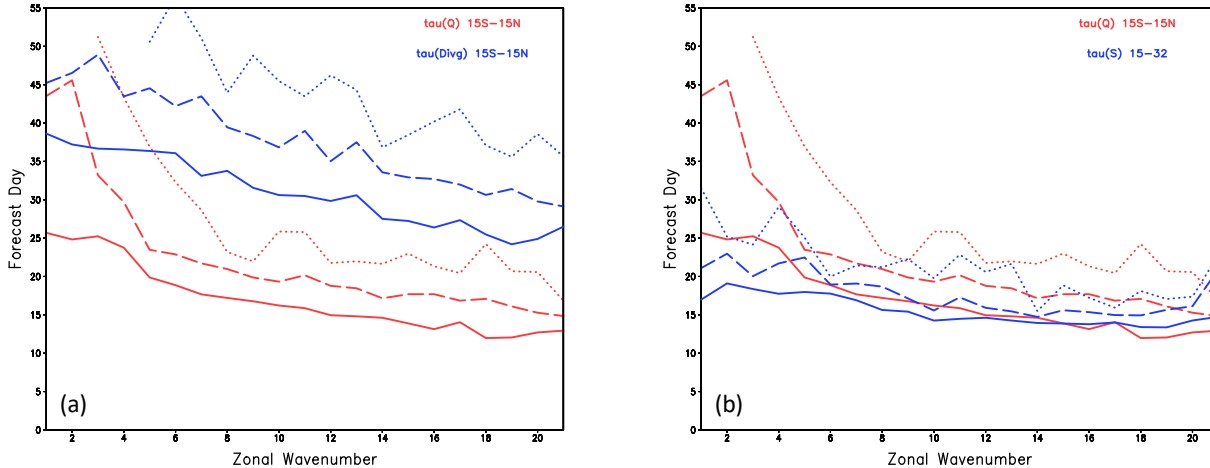

**Figure 4.** (a) The forecast day $\tau$ at which the error variance of heating $Q$ (averaged $15^o$ S - $15^o$N) reaches a fraction $f_\tau$ of the external error for $f_\tau$ = 0.50. 0.70 and 0.90 shown in red solid, dashed and dotted lines. The blue lines show $\tau$ for the 200 hPa divergence averaged over the band $15^o$ S - $15^o$N. (b) $\tau$ for the error variance in heating as in (a) along with $\tau$ for the Rossby wave source ($S$) averaged over $15^o$N - $32^o$N shown in blue lines.

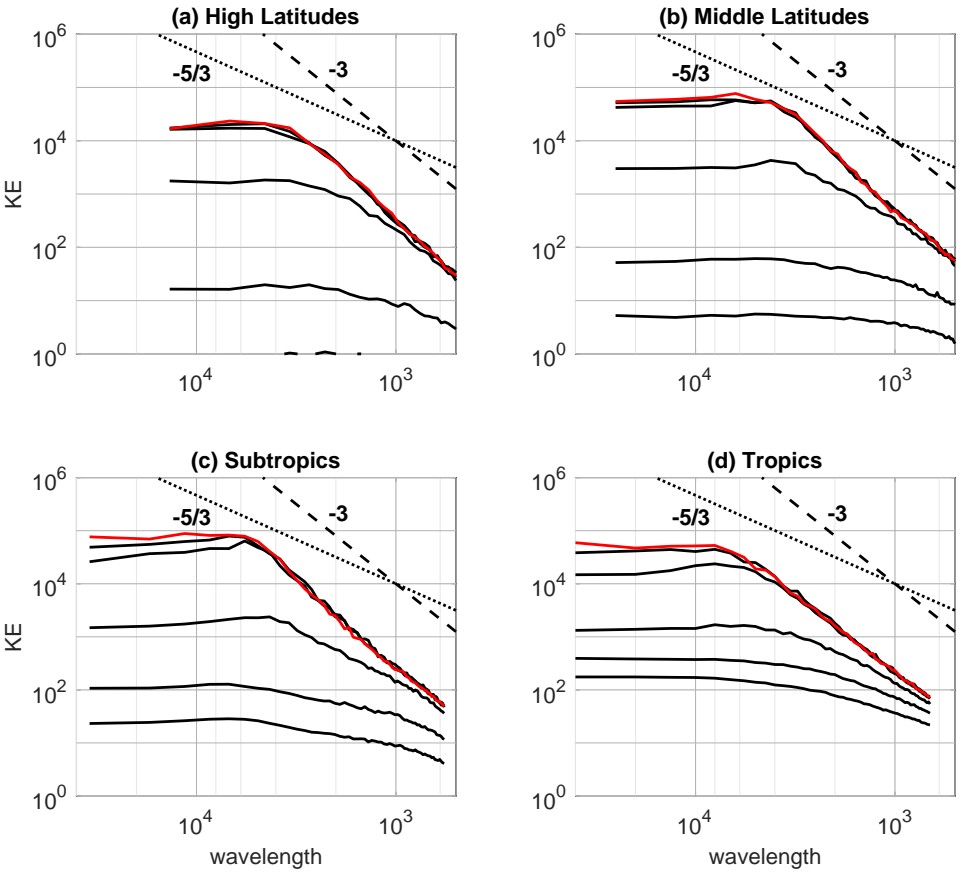

**Figure 5.** (a) Error variance spectra of 300 hPa kinetic energy, averaged over High Latitudes ($65^o$ N - $75^o$N), shown at forecast days 3, 5, 10, 20, 30 and 60 (with the 60 day error shown in red). (b) As in(a) but for Middle Latitudes ($45^o$ N - $55^o$N). (c) as in (a) but for the Subtropics ($25^o$ N - $35^o$N). (d) as in (a) but for the Tropics ($15^o$ S - $15^o$N). Units of $log_{10}(m^2 s^{-2})$. Reference spectral slopes corresponding to a $k^{-5/3}$ and $k^{-3}$ dependence are indicated.

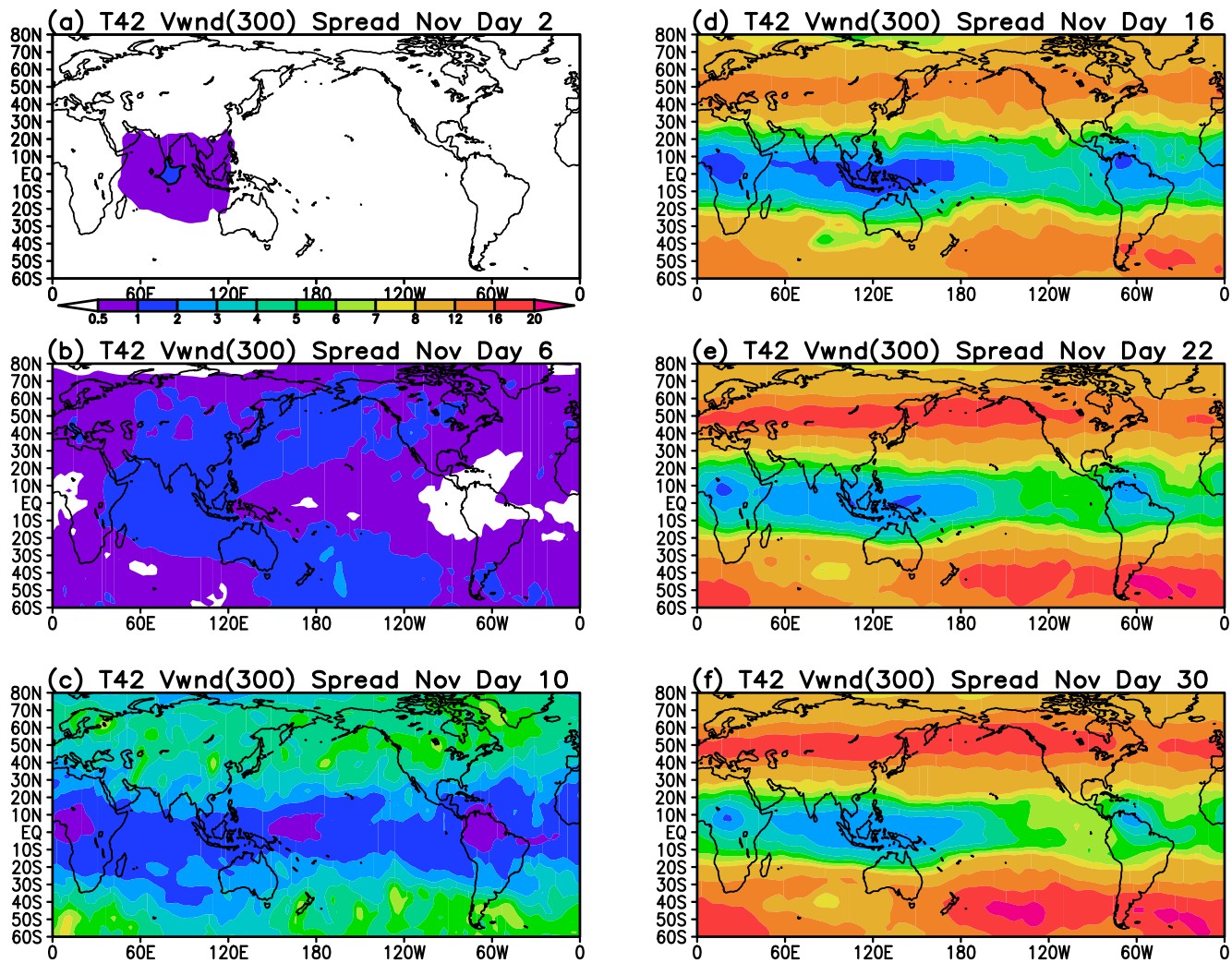

**Figure 6.** Ensemble spread of meridional wind $v$ at 300 hPa for forecast days 2, 6, 10, 16, 22, 30. Units of $ms^{-1}$.

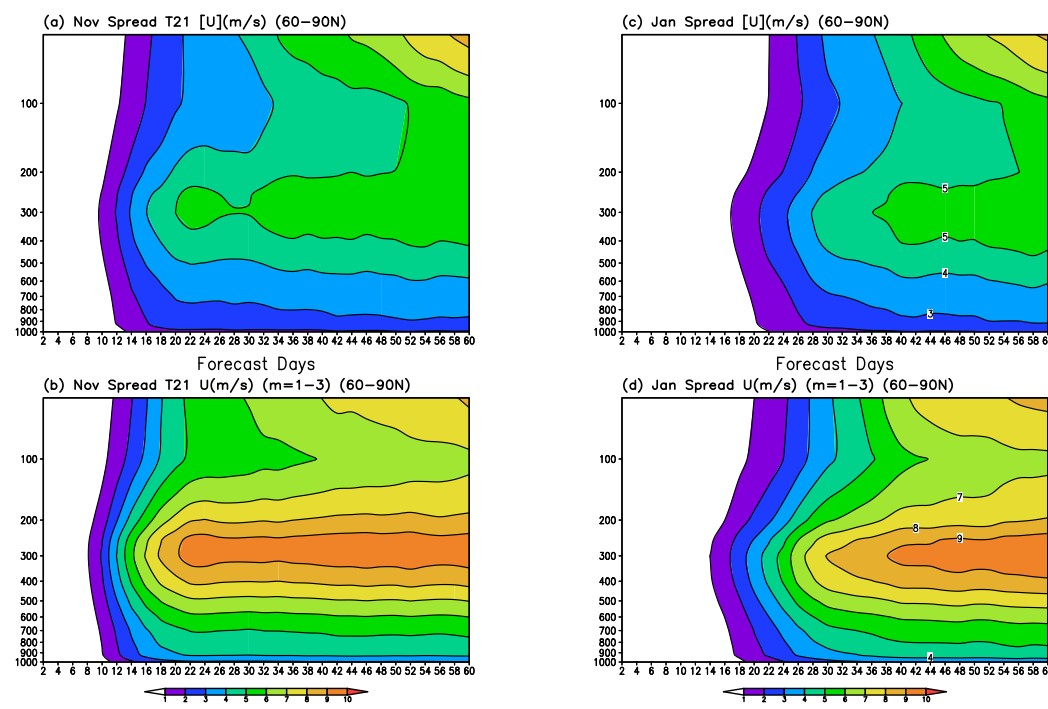

**Figure 7.** Ensemble spread of the zonal wind $u$, averaged 60N-90N for all levels (1000 - 50 hPa). The spread of the zonal-mean wind $[u]$ is shown in panels (a) and (c), and the spread due to zonal wavenumber 1-3 (panels (b) and (d)). The average spread over all November experiments is shown in panels (a) and (b), over all January experiments in panels (c) and (d). Units of $ms^{-1}$.)

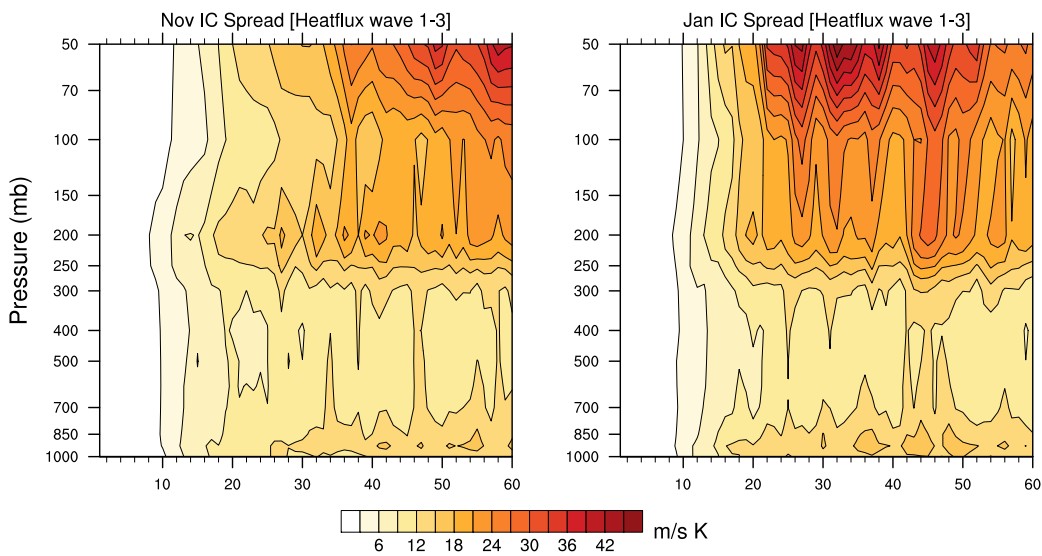

**Figure 8.** Ensemble spread for meridional eddy heat flux (summed over zonal wavenumbers 1 to 3) averaged between 40-80°N for November (left) and January (right) experiments. Units of $ms^{-1}K$.

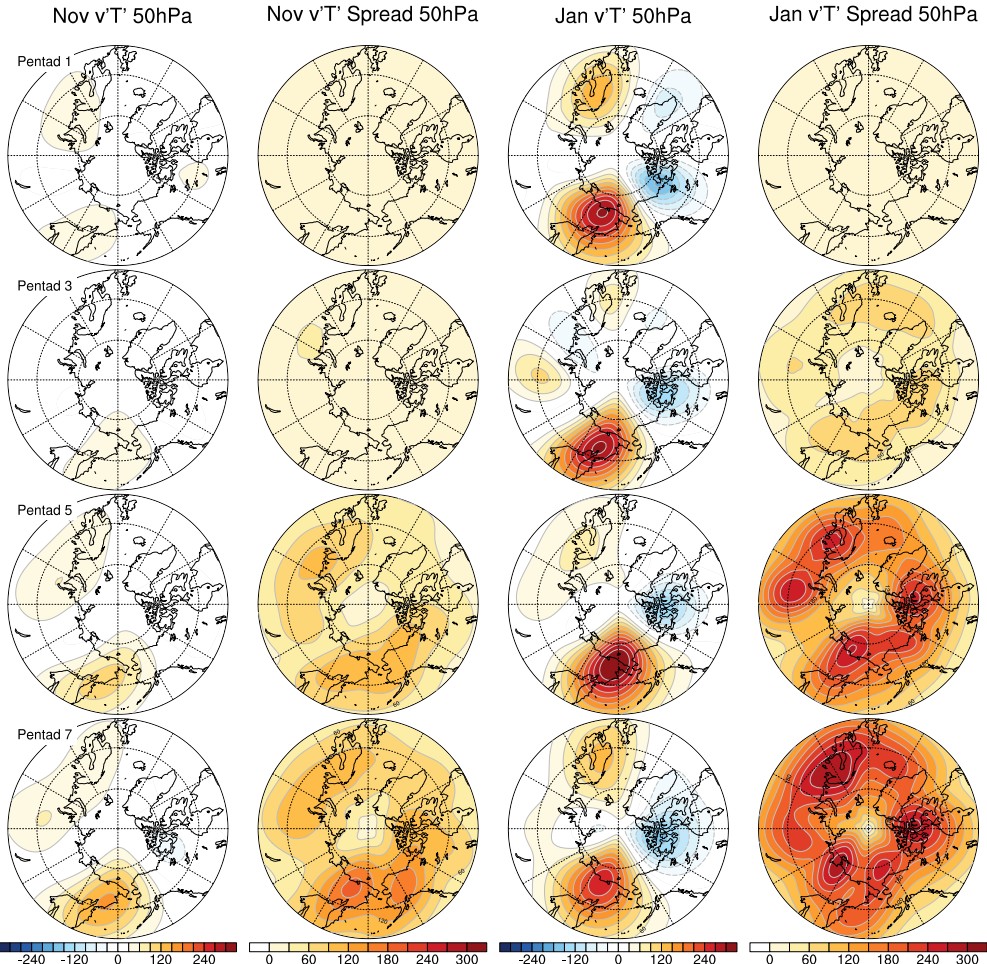

**Figure 9.** Geographical distribution of the planetary wave contribution to the zonal mean ensemble average and ensemble spread of meridional eddy heat flux at 50hPa. The ensemble average is shown in columns 1 and 3, the ensemble spread in columns 2 and 4 (as labeled). Rows 1 - 4 show averages over days 1-5, 11-15, 21-25 and 31-35 respectively. November experiments are given in columns 1 and 2, January results in columns 3 and 4. Contour interval is 30 $ms^{-1}K$.

## 6 Appendix

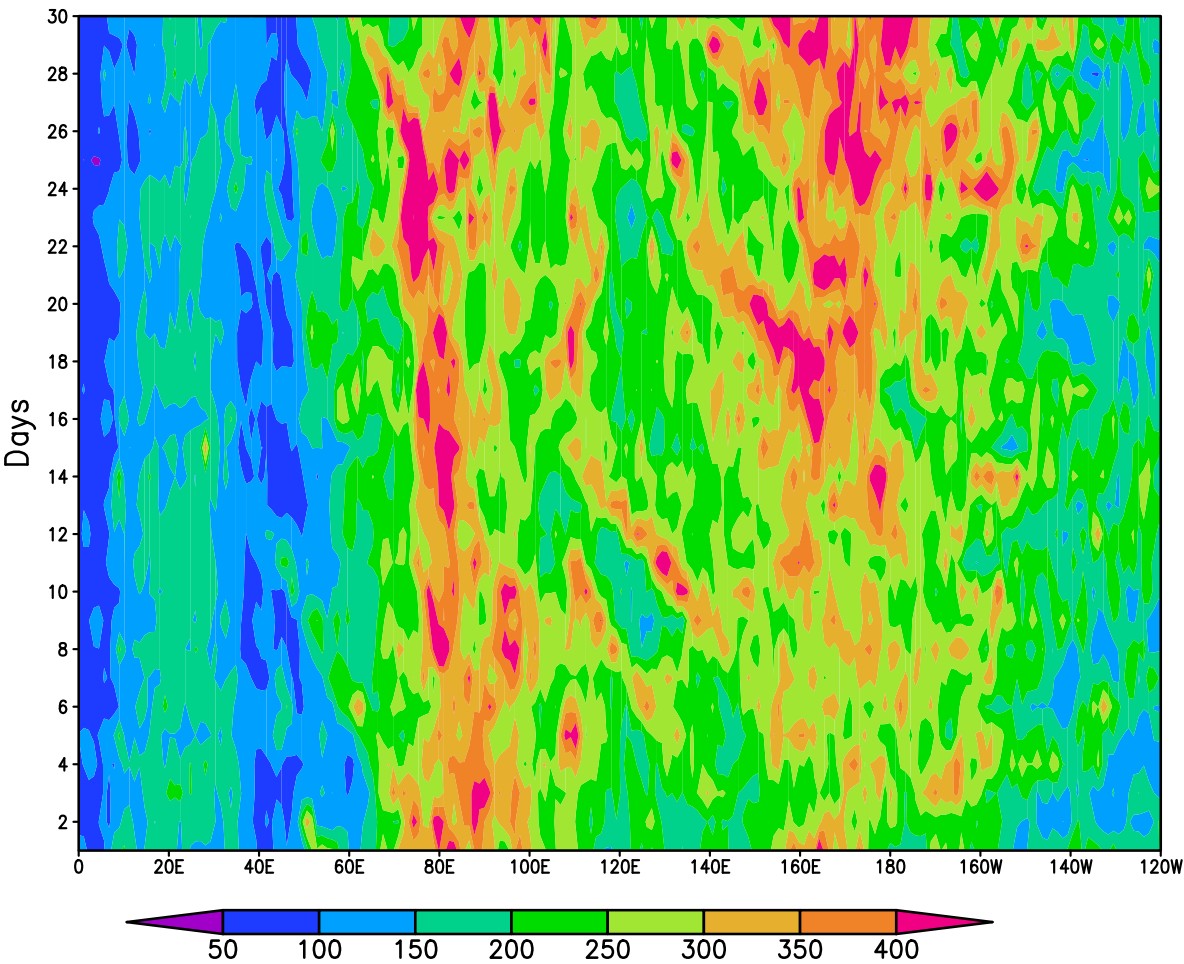

**Figure A1.** Variance of daily mean, vertically integrated diabatic heating (estimated from ERA5 reanalyses for days 1Nov to 30Nov) from the eight years corresponding to the Nov. experiments. See text for details. Units of $Wm^{-2}$.

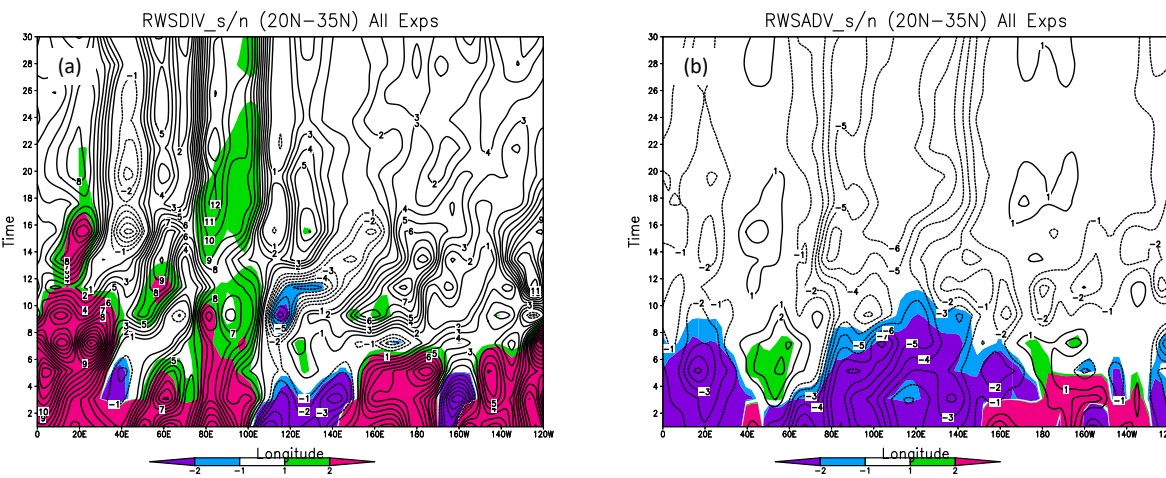

**Figure A2.** Evolution of the two components of the Rossby Wave Source ($S$): (a) the stretching term and (b) the advection term (as in Equation 1), averaged over all experiments. The terms were computed at the equivalent of T21 triangular spectral truncation (see text for details). $S$ was averaged between $20^{o}N$ and $35^{o}N$. The color scale gives the ratio of the ensemble mean to ensemble spread. The units of the RWS are $ms^{-1}$.

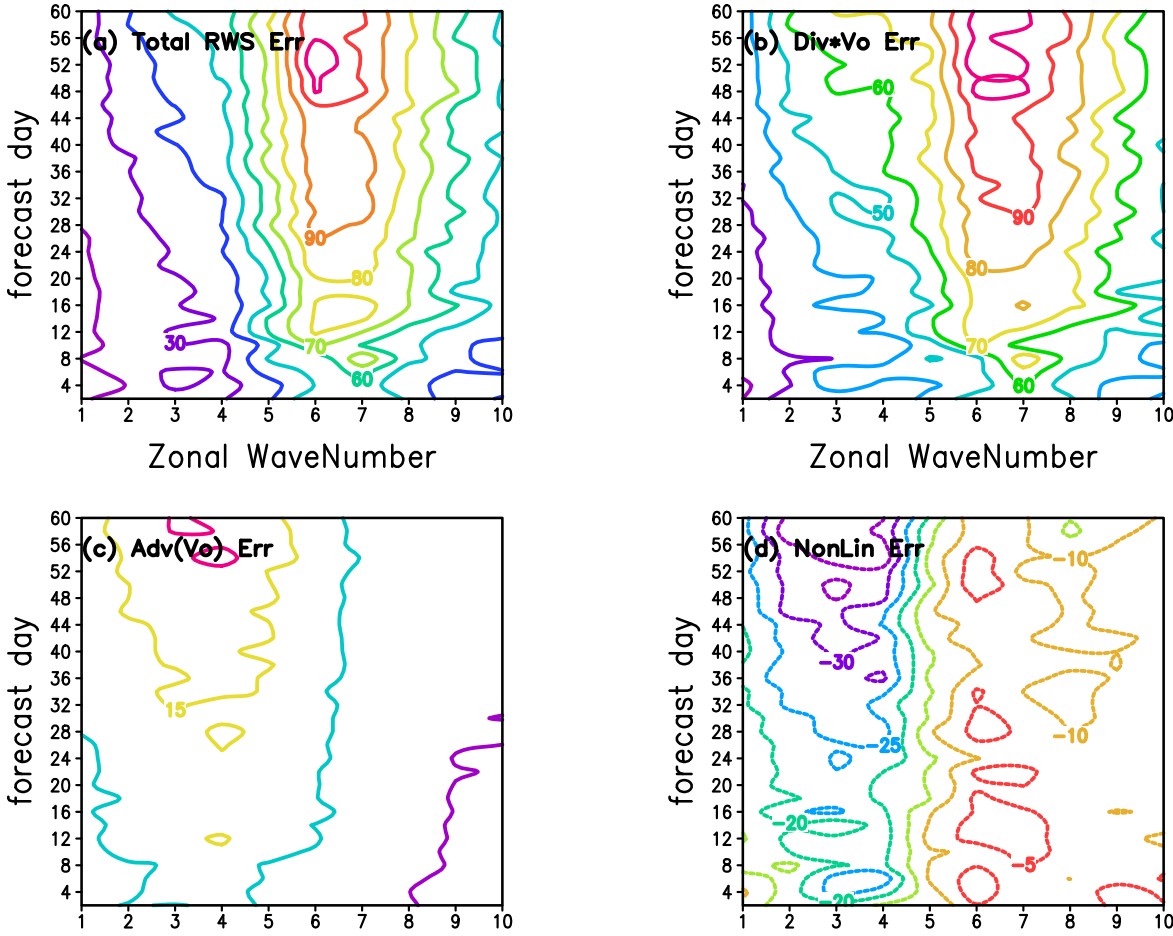

**Figure A3.** Evolution of the ensemble error of the Rossby Wave Source ($S$): (a) the total ensemble error $V_{ens}^{(RWS)}$; (b) the error due to the stretching term alone; (c) the error due to the advection term alone; (d) the error due to the interaction of the two terms, obtained by subtracting the two contributions (b) and (c) from the total. The errors are averaged over the latitude band $20^{o}N - 35^{o}N$. For details see equation 1) and the text. All results are averaged over all experiments. The units of the RWS error are $(m^{2}s^{-2})$.

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
