# Peer review of "Intrinsic Predictability Limits arising from Indian Ocean MJO Heating: Effects on tropical and extratropical teleconnections"

_EGUsphere, 2023_

## Referee Comment (RC2)

Review of egusphere-2023-493

Summary:
This is an interesting and generally well-written study about atmospheric predictability. The authors investigate the predictability limits that arise due to uncertainty in the diabatic heating of MJO convection. The experiment is based on the IFS model and uses a "perfect model" approach. The results suggest that the predictability limits (as defined by error saturation) for zonal wavenumbers 0-3 in the tropics are >60 days, 30-40 days for wavenumbers 4-10, and 20-30 days wavenumbers 11-21. The error, which is initially confined to the tropics, contaminates the extratropics via the mid-latitude storm tracks at around day 9-10. There is some discussion about Rossby wave source and downward propagation of error in the stratosphere, but it is less clear how those results fit into the narrative of the manuscript.

Recommendation:
Major revision

Paper Strengths:
The experiment set up is clean and so are the results, especially the timeseries plots of error growth. A noteworthy result is that the large scales in the tropics have longer predictability than often assumed, in other words there is untapped predictability potential in the tropics.

Major Comments:
1. While it is true that only the model physics over the Indian Ocean are perturbed, the effect of chaos seeding (Ancell et al. 2018) quickly spreads the error over the whole globe in non-physical ways. This likely means that the results are indistinguishable from results that would have been obtained if SPPT perturbations were added to the entire tropical belt or to another location that is convectively active, such as the Amazon. I suggest testing this out.
2. It would be interesting to see how different the result would be when "butterfly seeding" were used, i.e., tiny initial perturbations of the initial conditions everywhere on the globe, as in Judt (2018) or Zhang et al (2019). I recommend running an additional ensemble with this kind of perturbation and comparing the results with the ones obtained so far (this additional experiment would also be more in line with *intrinsic predictability*, which usually addresses predictability limits arising due to miniscule initial condition uncertainty).
3. Is it necessary to discuss the Nov and Jan initialization experiments separately? In my opinion no, as the differences between Nov and Jan events are not large enough to warrant the extra work for the reader to keep track of two sets of results. I therefore recommend combining all experiments into one "grand experiment". This shouldn't affect the conclusions.
4. It looks like Fig. 3 is not referenced in the text. Furthermore, I am not sure why the "Rossby wave source" is analyzed at all. I suggest removing this analysis or better motivating it.
5. Section 3.4 seems to be lacking a conclusion, or at least I'm left with this impression.

Minor Comments:

1. L.45: How does the presence of baroclinic instability limit the predictability of MJO teleconnections? Through error growth associated with baroclinic instability?
2. Figs. 1-3 and 6-9 imply that the runs are 30 days long, while the other figures show the entire 60 day time period. Why are only the first 30 days shown in the Hovmöller plots? Maybe nothing interesting happens after 30 days, but then it should be indicated somewhere so the reader doesn't end up confused whether or not the experiments are 30 or 60 days long.
3. L.102: Just curious, why are you not using ERA5 to initialize the ensembles?
4. L. 155-177: I don't think the description of the figures is necessarily wrong, but I do see a lot of noise in the Hovmöllers and not so much of the described
5. Fig. 1 (and Fig. 2): The evolution of the standard deviation in panels (d) shows very little propagation with the maximum being anchored between 70 and 100 deg E, unlike the heating amplitude . Is this because of the continuous perturbation in this region?
6. L. 188: "In the Indian Ocean the two fields are comparable." I disagree, there seems to be more red in Fig. 1d than in Fig. A1 over the Indian Ocean.
7. L. 211: Where does the 0.5 threshold come from? It seems arbitrary.
8. L. 298: "small scales" is relative, wavenumber 21 is "large scale" from the view of a synoptic/mesoscale meteorologist.

Editorial Suggestions:

1. L. 186: two "the"

References:

Ancell, B. C., A. Bogusz, M. J. Lauridsen, and C. J. Nauert, 2018: Seeding Chaos: The Dire Consequences of Numerical Noise in NWP Perturbation Experiments. Bull. Amer. Meteor. Soc., 99(3), 615-628, https://doi.org/10.1175/BAMS-D-17-0129.1.

Judt, F., 2018: Insights into Atmospheric Predictability through Global Convection-Permitting Model Simulations. J. Atmos. Sci., 75, 1477–1497, https://doi.org/10.1175/JAS-D-17-0343.1.

Zhang, F., Y. Q. Sun, L. Magnusson, R. Buizza, S. Lin, J. Chen, and K. Emanuel, 2019: What Is the Predictability Limit of Midlatitude Weather?. J. Atmos. Sci., 76, 1077–1091, https://doi.org/10.1175/JAS-D-18-0269.1.

---

## Author Response (AR1)

Response to Reviewer R1

The authors appreciated the comments of the reviewer and have substantially re-written the paper to address them.

The paper deals with limits of predictability associated with uncertainties in simulated diabatic heating over the Indian ocean during the MJO. This is investigated by re-running a set of 60-day long ensemble forecasts assuming perfect initial conditions. In other words, the study addresses a component of predictability associated with applied model errors. The error was simulated by stochastic perturbations of the tendencies due to physics geographically limited to the Indian Ocean region 50E-120E and 20N-20S. Their effects on predictability are studied in the ensemble spread of the vertically integrated diabatic heating, of Rossby wave source, of the heat flux and of the vorticity field at 200 hPa. The authors report predictability limits, defined as 0.5 of the saturation value of the ensemble variance, to be between 2 and 3 weeks.

We have a subtly different interpretation of our experiments, and have tried to clarify this in the revision. Our interpretation is not that our study addresses that component due to intrinsic variability in the model ("internal error") which mimics the uncertainty of heating even with a given MJO episode. The stochastic parameterization scheme which is used (SPPT) is considered an integral part of the IFS, and other research (i.e. Selz, 2019) supports the notion that the use of such schemes gives more realistic estimates of intrinsic predictability. The novelty of our experiments is that we suppress SPPT outside the Indo-Pacific region, and that is the only change we make. See the following changes where we try to make this more clear:
lines 50-52,
"However, another limitation to predictability is the variability of heating among different observed episodes of a given phase, and the intermittency of heating and other sub-grid scale processes in space and time even within a particular episode."
line 64
"The SPPT alters the instantaneous tendencies of the temperature, specific humidity and horizontal wind components due to sub-grid scale physics processes by scaling these tendencies up or down in a stochastic manner. It is considered a standard component of the IFS."
lines 344-345
"The suite of ensemble reforecast experiments presented here was explicitly designed to gauge the effect of the intrinsic un certainty of sub-grid motions on the response to the MJO in phases 2 and 3."

Summary:
The authors address an important problem but the paper in its current shape does not provide significant new insights on the teleconnections associated with the MJO or associated predictability limits.

While a number of papers have discussed the role of initial condition uncertainty (see next response), but the role of the uncertainty in heating has not been discussed as much. Lines 53-58:
"The dynamical character of the uncertainty in the response to this intermittency has been studied by Kosovelj et al. (2019) in a low resolution global model using idealized stochastic parameterizations. Similarly, the response to model systematic error in Indian Ocean temperatures was addressed by Zhao et al. (2023). The purpose of this paper is to extend the work of Kosovelj et al. (2019) to address the intrinsic limits of potential predictability due to the intermittency of heating and other sub-grid scale processes in a high resolution operational forecast model setting."
In the original manuscript we did not emphasize the predictability of the Rossby wave source in the paper enough. the rate of spread of the Rossby wave source gives an indication of how the spread of the tropical heating is translated into the uncertainty in the mid-latitude response.
In the revised version, see lines 236 – 241:
"…the corresponding [predictability] times for the Rossby wave source, shown in the blue curves in Figure 4b, are considerably shorter than those for Q. This is especially true for the planetary waves (wavenumbers 1 - 3) for which the predictability time for the RWS is shorter than that for the heating … This reflects the sensitivity of the RWS to the sub-tropical divergent flow and also the sub-tropical absolute vorticity (as expressed in equation 1). The predictability times for the RWS have not, to our knowledge, been shown before, and are an important result regarding the predictability of the extra-tropical circulation."

"A naive interpretation of the tropical divergence as the main forcing function for the extra-tropics would indicate long-range predictability related to the MJO. However, the predictability times for the RossbyWave Source S are considerably shorter: the largest scales reach 0.50 (0.70) of saturation already at around 20 (30) days. This is understandable since S is influenced not only by tropical and subtropical divergence but also by the meridional gradient of the jet. One path by which mid-latitude and sub-tropical variability may affect the response to tropical forcing is by changing the effective source for that response."

One way to revise the paper would be to compare the results with the operational seasonal forecasts which sample uncertainties in both initial conditions and model errors.

In the revised manuscript we differentiate between initial condition uncertainty and the uncertainty in sub-grid scale processes such as heating. The point is that error growth due to ICs can be tuned by changing the magnitude of the IC errors, where as the strength of the SPPT perturbations are an integral part of the IFS model used for operational forecasting.

"All members of each ensemble use the identical initial conditions, so that the only cause of model uncertainty (also called error here) must be the noise in heating introduced by the SPPT in the tropical Indian Ocean. We should point out here that one might design a similar set of experiments in which only the initial conditions were perturbed. After all, any perturbation whatever to the system will quickly propagate and grow (Ancell et al., 2018). The question is how quickly such errors grow and saturate, and the answer depends on, among other factors, the amplitude of the initial errors. In fact, Zhang et al. (2019) use the this dependence of rate of growth on the magnitude of the initial condition error to estimate the predictability limit of mid-latitude weather."

Detailed comments:

1. Assumptions

The authors argue (Lines 55-56) that they apply the "perfect model" assumption in their study of (Lines 46-47) "uncertainties in MJO heating, as witnessed by the variability in the details of heating among different episodes of a given phase." The statement should be re-written in line with what has been done (perfect initial-conditions and simulated model error). Please discuss what is meant by "variability in the details of MJO heating" and provide references.

We agree that the original manuscript was not clear. Here are our clarifications and additions:

"However, another limitation to predictability is the variability of heating among different observed episodes of a given phase, and the intermittency of heating and other sub-grid scale processes in space and time even within a particular episode. The dynamical character of the uncertainty in the response to this intermittency has been studied by Kosovelj et al. (2019) in a low resolution global model using idealized stochastic parameterizations. Similarly, the response to model systematic error in Indian Ocean temperatures was addressed by Zhao et al. (2023).

The purpose of this paper is to extend the work of Kosovelj et al. (2019) to address the intrinsic limits of potential predictability due to the intermittency of heating and other sub-grid scale processes in a high resolution operational forecast model setting. In this paper we do not address model systematic error."

"The suite of ensemble reforecast experiments presented here was explicitly designed to gauge the effect of the intrinsic uncertainty of sub-grid motions on the response to the MJO in phases 2 and 3. Each ensemble has all its members initialized identically during an observed MJO event, and differ from each other only in the realization of the stochastic parameterizations, applied only in the tropical Indo-Pacific region. Thus even though the errors (deviations within the ensemble) spread globally, they are ultimately due to the uncertainty in this region."

2. Methodology

a) Lines 59-61: "the stochastic parametrization scheme (SPPT) described in Leutbecher et al. (2017) has been altered so that perturbations which affect (directly or indirectly) diabatic heating tendencies are confined to the tropical Indian Ocean region". Some details would be useful here,

such as the amplitude of perturbations compared to the signal. Different wording on what parts of the model physics have been perturbed is provided at different places in the paper, and it should be clarified.

We try to be more consistent in referring to the SPPT as perturbing (adding a stochastic component to) the sub-grid scale physics in general.

lines 60-65:

"…the stochastic parametrization scheme (SPPT) described in Leutbecher et al. (2017) has been altered so that perturbations which affect (directly or indirectly) diabatic heating tendencies are confined to the tropical Indian Ocean region. The SPPT alters the instantaneous tendencies of the temperature, specific humidity and horizontal wind components due to sub-grid scale physics processes by scaling these tendencies up or down in a stochastic manner. It is considered a standard component of the IFS. The physics processes include those due to turbulent diffusion and sub-grid orography, convection, cloudiness and precipitation, and radiation."

lines 344-345 (in the Conclusions);

"The suite of ensemble reforecast experiments presented here was explicitly designed to gauge the effect of the intrinsic un certainty of sub-grid motions on the response to the MJO in phases 2 and 3."

b) The computation of the divergent horizontal wind in Eq. (1) should be explained.

Agreed. Lines 144 -162 give an fairly detailed outline of the mathematical steps used to compute the Rossby Wave Source. We make use of the transforms of a vector field to the corresponding divergence and curl in spherical harmonic space, and their inverse. These transforms are applied in several ways. We do not reproduce the details in this response; please see the lines indicated above.

c) What is the sensitivity of the results to the choice of the latitude belt used to compute S?

We discuss this sensitivity in ines 164-167: "…we consider the average source between 15N and 3N. The … The results shown in this paper are robust to changes in the latitude band chosen, both to modest poleward displacement and to widening it by 5 degrees."

The new Figure 2 shows the sensitivity of the mean and standard deviation of the Rossby Wave Source to different latitude bands. Also, the new Figure 4, showing the predictability times $\tau$ of the RWS as a function on zonal wave number (for different definitions of $\tau$) is remarkably robust to modest shifts and widening of the latitude band used.

d) Why is the integrated diabatic heating a good measure of the MJO predictability as compared for example with precipitation?

The focus of the paper is on how uncertainty in sub-grid scale processes in the tropics affect the extra-tropical predictability, and not on tropical precipitation. Within the tropics, it is the total heating that forces the atmospheric divergence and Rossby Wave Source.

lines 79-80:

"Since our goal is to document both the uncertainty in the tropical heating and the mid-latitude response in these experiments, we also consider the pathway by which the tropical heating forces extratropical Rossby waves."

lines 348-350:

"These subsequent errors in the tropical diabatic heating, tropical upper-level divergence and Rossby Wave Source indicate the path towards mid-latitude uncertainty in the circulation response."

However, we understand that in terms of extra-tropical spread of uncertainty, the heating is not the usual field used. We have replaced the maps showing the spread of ensemble spread with a single figure showing the evolution of ensemble spread of the meridional wind (new Figure 6).

Lines 253-255:

"To get a sense of how the errors spread geographically, we present maps of the ensemble spread of the meridional wind in Fig. 6. The choice of meridional wind was motivated by its close relationship with storm tracks and circumpolar wave guide (Branstator and Teng, 2017)."

e) Why is vorticity a good measure of the forecast error growth in the tropics (Figures 8-9)?

We have removed the maps of ensemble spread of vorticity in the tropics (see above response).

f) How is the estimated predictability limit sensitive to different choices of predictability time (Line 211) taken to be 50% of the saturation error?

Following Judt (2020) we show predictability times for different thresholds (50%, 70%, 90% of saturation) in the new Figure 4.

Lines 227-228:

"…we show the time τ at which the error variance of Q reaches a fraction f of the variance of the external error for f = 0.50 , 0.70, and 0.90, as a function of zonal wavenumber in Figure 4a. …"

g) How is the wavenumber analysis performed, is it spherical harmonics space?

The zonal wavenumber analysis is performed on latitude circles (i.e. as a function of longitude only) after the fields have been transformed back to the Gaussian grid.

h) The ENSO events are introduced in 2.2 and Figures 1-2, but little mentioned after 3.1.

We have added, in the Discussion (lines 289-293):

"The evolution of tropical heating shown in Figures 1 for El-Niño years shows less eastward propagation from the Indian Ocean compare to normal years, in line with the 290 findings of Liu et al. (2020), likely because less moisture is available over the Indian Ocean due to the ENSO convection in the central Pacific. Nevertheless, the average over all forecasts does show distinct eastward propagation for the first 10 days or so."

3. Relation to previous work

Many studies addressed the response of tropical and extratropical circulation to MJO-like heating perturbations. I disagree with the authors' statement that (Lines 47-49) "the wealth of MJO teleconnection research discussed above has relied almost exclusively on the Wheeler-Hendon multivariate empirical orthogonal function framework (Wheeler and Hendon, 2004)." See for example https://doi.org/10.1175/JAS-D-18-0203.1 and references herein. Similarly, there is a wealth of research in predictability associated with MJO that is missing in the introduction and discussion of the results.

We respond to this point and the next one together. In order to compare to previous work on evolution of error spectra due to initial condition or heating uncertainty, we have produced two new figures. Figure 3 shows the evolution of the spectra of mid-level heating, and Figure 5 shows the evolution of the error spectra of kinetic energy for various latitude bands. We present and discuss these new figures in the context of previous work.

In the Introduction:
lines 52-57:

"The dynamical character of the uncertainty in the response to this intermittency has been studied by Kosovelj et al. (2019) in a low resolution global model using idealized stochastic parameterizations. Similarly, the response to model systematic error in Indian Ocean temperatures was addressed by Zhao et al. (2023).

[revised manuscript text omitted]

4. Results
This paper, like several earlier studies of predictability, finds a predictability scale of 2-3 weeks and that longer scale have longer predictability. In the present study, the predictability limit is due to perturbations in model physics. Previous studies such as https://doi.org/10.1175/JAS-D-19-0116.1 find similar intrinsic predictability to be due to small perturbation in initial conditions (perfect model assumption). I wish the authors discussion their predictability results in comparison to other studies of predictability in the tropics and globally.
How does the growth of spread in selected variables compare with the scale-dependent circulation response to heating perturbations in  https://doi.org/10.1175/JAS-D-18-0203.1 (their figures 7-8)?
Please see detailed response to previous query.

Overall it is unclear why the ensemble spread of diabatic heating is a good measure of predictability limits, rather than prognostic variables of circulation and/or precipitation.  For example, how is the amplitude of the ensemble spread in diabatic heating related to the predictability of precipitation? Could the results be coupled with the precipitation validation in the ECMWF model forecasts (e.g. https://doi.org/10.1029/2020GL091022)?
In the tropics, the spread of diabatic heating is presented to document just how variable heating is, and to show the forcing for the extra-tropical spread in circulation. Whether the tropical precipitation in this model is realistic is a different, albeit interesting question.

Replies to R2

We appreciate the comments of the reviewer, and have done our best to address them. The paper has been rather extensively modified.

1. While it is true that only the model physics over the Indian Ocean are perturbed, the effect of chaos seeding (Ancell et al. 2018) quickly spreads the error over the whole globe in non-physical ways. This likely means that the results are indistinguishable from results that would have been obtained if SPPT perturbations were added to the entire tropical belt or to another location that is convectively active, such as the Amazon. I suggest testing this out.

This is a valid point – any perturbation will rapidly spread due to variety of reasons, some numerical and some intrinsic to the system. We want to point out that the stochastic parameterization scheme which is used (SPPT) is considered an integral part of the IFS, and other research (i.e. Selz, 2019) supports the notion that the use of such schemes gives more realistic estimates of intrinsic predictability. Thus the effects of restricting the SPPT to the tropical Indo-Pacific region may be confined to the early growth of errors.

In the Introduction
lines 75-79:
" Another question regarding our experimental design is whether localizing the application of the SPTT to the tropical Indian Ocean is necessary, since following the argument of Ancell et al. (2018), perturbations in any region will quickly propagate to the Indian Ocean region. The only way to answer this question is to re-run the same set of experiments with SPPT applied globally, which is the subject of future research. We will return to this question in the Discussion section."

In the Discussion:
Lines 294-300:
"The evolution of the average of the ensemble spread in vertically integrated heating ( Q) shown in Figure 1d shows clearly that the within-ensemble variability induced by the application of the regionally confined SPPT remains mostly confined to that region ( 50 -120E) for the first 10 days or so. This is also true for the tropical meridional wind spread (Figure 6) for the first 6 days. This suggests that the evolution of the tropical heating and circulation uncertainties would be different had the SPPT been applied throughout the tropical belt. Whether this difference would strongly affect the growth of uncertainty in the extra-tropics is hard to assess directly from these experiments. This question awaits future research."

2. It would be interesting to see how different the result would be when "butterfly seeding" were used, i.e., tiny initial perturbations of the initial conditions everywhere on the globe, as in Judt (2018) or Zhang et al (2019). I recommend running an additional ensemble with this kind of perturbation and comparing the results with the ones obtained so far (this additional experiment would also be more in line with intrinsic predictability, which usually addresses predictability limits arising due to miniscule initial condition uncertainty).

While we agree that this would be an interesting comparison, there is an arbitrariness in perturbing initial conditions. In fact the whole purpose of the cited paper of Zhang et al. is to compare the error growth due to current, operational uncertainty in the initial conditions (taken from multiple reanalyses) to a hypothetical "perfect" scenario that is implemented by arbitrarily reducing the initial condition uncertainties by a factor of 100. Such a study could be implemented in the ECMWF modeling system since there is in place a procedure to perturb the initial conditions (although we did not use this in the simulations). As above, this computer-intensive proposal is a goal for a future project.

Lines 66-74:
"All members of each ensemble use the identical initial conditions, so that the only cause of model uncertainty (also called error here) must be the noise in heating introduced by the SPPT in the tropical Indian Ocean. We should point out here that one might design a similar set of experiments in which only the initial conditions were perturbed. After

all, any perturbation whatever to the system will quickly propagate and grow (Ancell et al., 2018). The question is how quickly such errors grow and saturate, and the answer depends on, among other factors, the amplitude of the initial errors. In fact, Zhang et al. (2019) use the this dependence of rate of growth on the magnitude of the initial condition error to estimate the predictability limit of mid-latitude weather. Judt (2020) study the dependence on the rate of initial condition error growth on region (tropical, mid-latitude and high-latitude) for short simulations using a global storm-resolving model.

3. Is it necessary to discuss the Nov and Jan initialization experiments separately? In my opinion no, as the differences between Nov and Jan events are not large enough to warrant the extra work for the reader to keep track of two sets of results. I therefore recommend combining all experiments into one "grand experiment". This shouldn't affect the conclusions.

This is a valid point for most of the diagnostics. It is only for the figures relating to the role of the stratosphere  that we have reasons to discriminate the Jan. and Nov. initial conditions. This is because the polar vortex may not be fully formed in November, so that the "stratospheric pathway" towards uncertainty growth will have a different time scale for the Nov and Jan runs.

Except for Figures 7, 8 and 9 relating to the stratosphere, all other diagnostics have the Nov and Jan experiments combined. Note that we have modified some of figures and added a number of new ones in order to address the concerns of another reviewer.

4. It looks like Fig. 3 is not referenced in the text. Furthermore, I am not sure why the "Rossby wave source" is analyzed at all. I suggest removing this analysis or better motivating it.

One of the new results in this paper is the predictability time of the Rossby wave source, so we have better motivated it:
In the Introduction:
lines 79-83:
 "Since our goal is to document both the uncertainty in the tropical heating and the mid-latitude response in these experiments, we also consider the pathway by which the tropical heating forces extratropical Rossby waves. Although the MJO-related tropical heating is expected to force a corresponding signal in upper tropospheric divergence, this signal generally occurs within an easterly background wind, where stationary Rossby waves are not expected to propagate. Sardeshmukh and Hoskins (1988) derive a more complete formulation of the source of barotropic Rossby waves (hereafter Rossby wave source…"

In the Results discussing the new Figure 4, which show wavenumber dependent predictability times for the tropical heating, upper-level tropical divergence and Rossby Wave Source. The times are shown to reach 50%, 70% and 905 of saturation.
lines 233-241:
"The predictability times for the T21 representation of the upper-level (200 hPa) divergence are shown in Figure 4b in the blue curves. These times are notably longer than for the vertically integrated heating. For example, the divergence [predictability time] corresponding to [70% of saturation] is greater than 35 days for the largest scales, compared to 20 days for the heating. While this might be taken
to indicate high predictability for the extra-tropical response, the corresponding times for the Rossby wave source, shown in the blue curves in Figure 4b, are considerably shorter than those for Q. This is especially true for the planetary waves (wavenumbers 1 - 3) for which the predictability time for the RWS is shorter than that for the heating by about 8 days [to reach 50% saturation} and by about 10 - 20 days for f  = 0:70. This reflects the sensitivity of the RWS to the sub-tropical divergent flow and also the sub-tropical absolute vorticity (as expressed in equation 1). The predictability times for the RWS have not, to our knowledge, been shown before, and are an important result regarding the predictability of the extra-tropical circulation."

In the Conclusions:
lines 345-350:
"Each ensemble has all its members initialized identically during an observed MJO event, and differ from each other only in the realization of the stochastic parameterizations, applied only in the tropical Indo-Pacific region. Thus even

though the errors (deviations within the ensemble) spread globally, they are ultimately due to the uncertainty in this region. These subsequent errors in the tropical diabatic heating, tropical upper-level divergence and Rossby Wave Source indicate the path towards mid-latitude uncertainty in the circulation response."

5. Section 3.4 seems to be lacking a conclusion, or at least I'm left with this impression.
In the Discussion:
Lines 340-352:
"The growth of uncertainty in the stratospheric circulation, as seen in Figure 7, is forced by the upward propagation of the planetary wave meridional flux of sensible heat (which is the dominant term in the vertical component of the Eliassen-Palm flux), shown in Figure 8. This uncertainty then propagates downward into the upper and middle troposphere. While most of the upper troposphere sensible heat flux is due to planetary wave disturbances in the Pacific, its uncertainty in the North Atlantic and Asian sectors are also large, especially for the Jan. experiments (Figure 9). This downward propagation is potentially linked to wave-mean flow interaction which acts to bring anomalies in e.g. wind and temperature to the lower stratosphere.
The planetary wave error in the upper troposphere (300hPa) for Nov. reaches a maximum 20 days earlier than does the error at 50hPa, hinting at a tropospheric forcing of the stratospheric spread. The stratospheric descent of error seen in Figure 7 occurs towards the end of the experiments, consistent with the tropospherically forced uncertainty being modulated by the stratospheric circulation (Domeisen et al., 2020b). This descent is seen about 10 days later in the reforecast period for the Nov. experiments than for the Jan. experiments. This is likely due to the lack of a fully formed stratospheric vortex during November, so that the establishment of a wave guide for vertically propagating (It was not possible to verify this since data were retained only up to 50hPa.)"

In the Conclusions:
lines 374-375:
"The role of the stratosphere in amplifying uncertainty is generally confined to the latter part of the 60-day reforecasts, after the ensemble spread in upper-tropospheric heat flux has affected levels above 50 hPa [Figures 7 and 8]."

Minor Comments:
1. L.45: How does the presence of baroclinic instability limit the predictability of MJO teleconnections? Through error growth associated with baroclinic instability?

Yes you are correct. But we have dropped the reference to baroclinic instability since it was apparently confusion.

2. Figs. 1-3 and 6-9 imply that the runs are 30 days long, while the other figures show the entire 60 day time period. Why are only the first 30 days shown in the Hovmöller plots? Maybe nothing interesting happens after 30 days, but then it should be indicated somewhere so the reader doesn't end up confused whether or not the experiments are 30 or 60 days long.

In the revised paper, Figures 1, 2 and 6 show only the first 30 days.

Reference to Figure 1:
lines 186-187
"The daily averaged evolution of vertically integrated diabatic heating anomaly is shown averaged for the first 30 days of the 60-day experiments in Figure 1…"

Reference to Figure 2 (where it is clear that the signal has dissipate by day 30):
lines 198-199
"figure 2 shows the evolution of both the ensemble average of the latitudinally averaged RWS, averaged over all experiments, for the first 30 forecast days."

Reference to Figure 6:
lines 253-258:
"…we present maps of the ensemble spread of the meridional wind in

Fig. 6. The choice of meridional wind was motivated by its close relationship with storm tracks and circumpolar wave guides (Branstator and Teng, 2017). Much of the tropics outside of the Indian Ocean region is nearly error free even at day 6. By day 10, substantial error has already appeared in the extra-tropics, particularly in the storm-track regions, and by day 16 the extra-tropical spread has almost saturated. By day 30 the spread in the extra-tropics has essentially reached its saturation value, since it doesn't increase for longer lead times (not shown)."

3. L.102: Just curious, why are you not using ERA5 to initialize the ensembles?
We purposely used the model configuration that ECMWF uses for its monthly forecasts, since that model has been well calibrated vis-à-vis the MJO. That configuration is set up to use ERA-Interim for initial conditions.

4. L. 155-177: I don't think the description of the figures is necessarily wrong, but I do see a lot of noise in the Hovmöllers and not so much of the described

Points 4 and 5 are addressed together, since they both relate to Figure 1, which now shows Nov and Jan experiments combined. The discussion has been clarified:
lines 186-193:
"The daily averaged evolution of vertically integrated diabatic heating anomaly is shown averaged for the first 30 days of the 60-day experiments in Figure 1c. The heating has been averaged over the tropical band (15S-15N), over all ensemble members and over all experiments. The eastward propagation of positive heating anomalies near longitude 90E can be seen for about 8 days, along with robust westward propagation of heating anomalies that appear after 4 days in the central Pacific. The ensemble spread of the heating (also averaged 190 over the tropical band and all experiments) is shown in Figure 1d. The dominant influence of the SPPT generated perturbations over the Indian Ocean sector is clear, leading to the largest ensemble spread in this sector.
In order to gauge the degree of influence of ENSO on the heating evolution, we also show the tropical heating anomalies separately for the six warm ENSO events (Nov 1986, Nov 1987, Nov 2002, Nov 2013, Jan 1987, Jan 2010) in Figure 1a and for the remaining seven experiments in Figure 1b. The warm events show the establishment of strong tropical heating in the central Pacific after day 15 (as expected), while both sets of years show robust westward propagation of the heating from the central Pacific. The initial eastward propagation of the Indian Ocean anomalies is somewhat delayed in the warm event years in comparison to the neutral years."

5. Fig. 1 (and Fig. 2): The evolution of the standard deviation in panels (d) shows very little propagation with the maximum being anchored between 70 and 100 deg E, unlike the heating amplitude . Is this because of the continuous perturbation in this region?
(see reply above)

6. L. 188: "In the Indian Ocean the two fields are comparable." I disagree, there seems to be more red in Fig. 1d than in Fig. A1 over the Indian Ocean.
We have refined the discussion.
lines 212-218:
"In order to determine whether the strength of the stochastic perturbations (reflected in the magnitude of the ensemble spread in the Indian Ocean region) is reasonable, we also computed the inter-annual standard deviation $\sigma_{IA}$ of the tropical Q from the ERA5 reanalysis over the eight years corresponding to the Nov. Experiment for the first 30 days. Figure A1 in the Appendix shows the daily evolution of the standard deviation with the same scale as in Figure 1d. The ERA5 $\sigma_{IA}$ is largely confined to the same regions as the model spread: the Indian Ocean region and the west-central Pacific. In the Indian Ocean sector, the model spread in heating has somewhat lower maximum values than $\sigma_{IA}$, but extends over a wider area. The model spread is notably less than $\sigma_{IA}$ over the Pacific up to forecast day 30."

7. L. 211: Where does the 0.5 threshold come from? It seems arbitrary.
        Both reviewers have made this point, and we have changed the analysis to include plots of wavenumber – dependent predictability for tropical heating, upper-level divergence and Rossby Wave Source for multiple thresholds: 0.50, 0.70 and 0.90. See the new Figure 4.
lines 227-241:

"To make this more precise, we show the time $\tau$ at which the error variance of Q reaches a fraction $f_\tau$ of the variance of the external error for $f_\tau = 0.50$, 0.70, and 0.90, as a function of zonal wavenumber in Figure 4a. The red curves give the results for Q in the solid, dashed and dotted lines, respectively. Prior to calculating the error variances for this plot, we have truncated Q (and the other fields to be shown) to a spherical harmonic T21 representation in order to eliminate excessive noise. $\tau$ increases with zonal scale (decreasing wavenumber) for all choices of $f_\tau$, but this is particularly noticeable for $f_\tau = 0.70$ and 0.90. In fact the limit of 0.90 of the external error is never reached for zonal wavenumbers 1 and 2.

The predictability times for the T21 representation of the upper-level (200 hPa) divergence are shown in Figure 4b in the blue curves. These times are notably longer than for the vertically integrated heating. For example, the divergence $\tau$ corresponding to $f_\tau = 0:70$ is greater than 35 days for the largest scales, compared to 20 days for the heating $\tau$. While this might be taken to indicate high predictability for the extra-tropical response, the corresponding times for the Rossby wave source, shown in the blue curves in Figure 4b, are considerably shorter than those for Q. This is especially true for the planetary waves (wavenumbers 1 - 3) for which the predictability time for the RWS is shorter than that for the heating by about 8 days for $f_\tau = 0.50$, and by about 10 - 20 days for $f_\tau = 0.70$. This reflects the sensitivity of the RWS to the sub-tropical divergent flow and also the sub-tropical absolute vorticity (as expressed in equation 1). The predictability times for the RWS have not, to our knowledge, been shown before, and are an important result regarding the predictability of the extra-tropical circulation."

8. L. 298: "small scales" is relative, wavenumber 21 is "large scale" from the view of a synoptic/mesoscale meteorologist.
Yes that was a bad choice of wording; we have omitted this phrase.

---

## Author Response (AR2)

**Report #1**

Submitted on 03 Jul 2023
Anonymous referee #2

**Anonymous during peer-review:** **Yes** No

**Anonymous in acknowledgements of published article:** **Yes** No

**Checklist for reviewers**

| | |
|---|---|
| **1) Scientific significance**
Does the manuscript represent a substantial contribution to scientific progress within the scope of this journal (substantial new concepts, ideas, methods, or data)? | Excellent **Good** Fair Poor |
| **2) Scientific quality**
Are the scientific approach and applied methods valid? Are the results discussed in an appropriate and balanced way (consideration of related work, including appropriate references)? | Excellent **Good** Fair Poor |
| **3) Presentation quality**
Are the scientific results and conclusions presented in a clear, concise, and well structured way (number and quality of figures/tables, appropriate use of English language)? | Excellent **Good** Fair Poor |

**For final publication, the manuscript should be**

accepted as is

**accepted subject to technical corrections**

accepted subject to minor revisions

reconsidered after major revisions

rejected

**Were a revised manuscript to be sent for another round of reviews:**

**I would be willing to review the revised manuscript.**

I would not be willing to review the revised manuscript.

**Suggestions for revision or reasons for rejection**
(visible to the public if the article is accepted and published)

The authors have made significant improvements to the manuscript, and their efforts are commendable. However, there is still some uncertainty regarding the authors' focus on perturbations applied only to the Indian Ocean region. I speculate here that similar results

would be obtained if perturbations were applied to other regions, such as the Amazon. Nonetheless, the authors acknowledge this limitation and suggest it as a subject for future research.

Note to the Reviewer: All line numbers refer to the pdf that includes all changes (both additions and deletions) denoted in red font.
Some minor comments:
L13: This zonal wave 1, the error variance of Q never reaches 90% of saturation. --- Something seems to be amiss in this sentence. This is fixed – see new L13.
L47: add the before mid-latitudes. This is fixed – see new L47
Fig. 3: How do we know that the spectrum doesn't grow after 60 days, i.e., that the red line is truly the saturations spectrum? The external error has been added to this Figure. See changes in new L248
L262: add space before the new sentence begins This has been done.

**Checklist for reviewers**

**1) Scientific significance**
Does the manuscript represent a substantial contribution to scientific progress within the scope of this journal (substantial new concepts, ideas, methods, or data)?

Excellent **Good** Fair Poor

**2) Scientific quality**
Are the scientific approach and applied methods valid? Are the results discussed in an appropriate and balanced way (consideration of related work, including appropriate references)?

Excellent **Good** Fair Poor

**3) Presentation quality**
Are the scientific results and conclusions presented in a clear, concise, and well structured way (number and quality of figures/tables, appropriate use of English language)?

Excellent Good **Fair** Poor

**For final publication, the manuscript should be**

accepted as is

accepted subject to technical corrections

**accepted subject to minor revisions**

reconsidered after major revisions

rejected

**Were a revised manuscript to be sent for another round of reviews:**

**I would be willing to review the revised manuscript.**

I would not be willing to review the revised manuscript.

**Suggestions for revision or reasons for rejection**
(visible to the public if the article is accepted and published)

Review of "Intrinsic Predictability Limits airising from Indian Ocean MJO Heating: Effects on tropical and extratropical teleconnections" by David M. Straus, Daniela I. V. Domeisen, Sarah-Jane Lock, Franco Molteni, and Priyanka Yadav

Note to the Reviewer: The questions/concerns led to an enhanced paper, for which we are grateful. All line numbers refer to the pdf that includes all changes (both additions and deletions) denoted in red font. More specific reponses are given below.

Synopsis:
The study by Straus et al. aims at identifying predictability limits arising from tropical diabatic heating over the Indian Ocean during MJO phases 2/3. For this purpose, a set of ECMWF-IFS

hindcasts is performed with the stochastic parametrization scheme (SPPT) being only applied in a limited range of longitudes over the tropics. The authors find that planetary wave components of the tropical heating and divergence is predictable out to 40 days. However, the Rossby wave source which allows the influence of the tropical heating to propagate to the extratropics is only predictable for 20-30 days. Except for numerous minor technical inaccuracies the paper is well written. In my view, the most important deficit is that the discussion lacks the role that systems other than the MJO could play. For example, the MJO modulates the storm track activity (Moore et al. 2010; Lee and Lim 2012) over the western North Pacific which effects the magnitude of the RWS and presumably its intrinsic predictability. After these minor comments have been addressed, I recommend the paper for publication in WCD.

- Since the paper only discusses error growth in phases 2-3, the storm track activity and hence Rossby wave source) dependence on the phase of the MJO does not affect our particular results. But it does mean that the RWS behavior we find is not universal, but only applied to MJO phases 2-3. Please see new L 392-394.

Minor:

l. 28: Do you mean the stratosphere with "upper atmosphere"? Yes – this has been fixed – see new L28

l. 56: "high resolution" is a relative term. Please try to be more specific. See added phrases on new L53 and new L56.

l. 69: It would be helpful to the reader if the motivation for such initial condition perturbation experiments was provided. Wouldn't such experiments also help to understand the effect of observational uncertainties in those regions? We added "Such experiments could be used, 70 for example to understand the potential impact of changes in the observing system on error growth." on new L69-70.

l. 101: You may want to add that the configuration is close to what was used at that time at ECMWF. (Not sure what is meant here?)

l. 112: Wang et al. (2023) report a pronounced seasonality of MJO teleconnections, e.g., with a stronger positive geopotential height anomaly over the central North Pacific in January to March compared to October to December. Thus, what is the motivation for choosing one initialization date in November and the other in January? Added on new L117-122: "The large ensemble size used dictated that we keep the number of MJO-phase 3 initial dates to a relatively small number (here 13). In order to make contact with previous reforecasts (the subject of a future publication), and to span varying parts of the seasonal cycle, 01 January and 01 November were chosen. However, we acknowledge that additional experiments would enable us to discriminate between the teleconnections in early and late winter, since they are known to

be different, see e.g. Abid et al. (2021). All reforecasts initialized on 01 January (01 November) will be referred to as Jan (Nov) reforecasts."

"

Table 2: Would it be possible for completeness to provide all Nino 3.4 indices in the table? Yes this has been done.

l. 174-178: Have you considered to provide the definitions of internal error variance, ensemble error variance, and external error variance as equations? This may make it easier to the reader to understand the three parameters. We have extensively revised the definitions of the different error variances, including detailed equations. See new L185-204.

l. 192: After having read the full manuscript I somehow wonder about the influence of ENSO. From my point of view, this discussion distracts from the main results of the study and I wonder if this aspect should be part of the paper also due to the limited sample size? We agree: Figure 1 now shows only the mean heating and its standard deviation averaged over all experiments, with no distinction between 01 November and 01 January forecasts. The discussion of the role of ENSO has been removed (new L214-219).

l. 201: It is interesting to note that the RWS maximizes roughly 10 days after forecast initialization and that the magnitude of the RWS is larger for the more northern latitudinal band. Is it possible that the high magnitude in RWS between 20 to 35°N is related to synoptic activity which typically increases after MJO phases 2 and 3 over the western North Pacific? . For example, Lee and Lim (2012) show negative Rossby wave source over the western North Pacific after MJO Phase 3 (their Fig. 3) which may be related to the outflow of rapidly ascending air streams in midlatitudes that typically reach their peak activity in the same region after MJO phases 2/3 (Quinting et al. 2023; their Fig. 1).
There are a number of revisions and new analyses that have been carried out in regard to the Rossby Wave Source, answering this concern and ones further in the list. We have:
- Replotted Figure 2 so that it shows the same longitude range as Figure 1
- In preparation for the discussion of the stretching and advection components of the RWS, we have expanded the equation and added a bit more description new L160-162.
- The evolution of the two components (stretching term and advection term) are now plotted in new Figure A2 (in the same format as Figure 2). A discussion has been added in new L 229-231.
- The contribution of the stretching and advection terms (and their interaction) as a function of zonal wavenumber and forecast time is plotted in new Figure A3, and discussed on new L 268-273.
- A discussion of the contribution of mid-latitude baroclinic systems to the RWS is added in new L344-348.

l. 202: The comparison of Figs. 1 and 2 is difficult due to the different longitude ranges.

Accordingly, I would like to kindly ask the authors to show the same longitude range in Fig. 1 as in Fig. 2.
- Fig. 2 has been replotted to use the same longitudes as Fig. 1

l. 239: This is related to my earlier comment: Does this indicate that the RWS is related to divergent flow of midlatitude disturbances rather than to divergent flow directly associated with the convection of the MJO itself? Further, the RWS includes divergence as well as the gradient of vorticity? Did you investigate separately for divergence and vorticity when the forecast error variance reaches a fraction of 0.5, 0.7, and 0.9?
- Please see the discussion of the Rossby Wave Source above. Since the RWS error variance includes (negative) terms resulting from the interaction between stretching and advection components, it would be difficult to interpret the predictability times for individual components.

l. 275: Though it is probably difficult to quantify, could the authors include some discussion on the seasonality of the MJO teleconnections. E.g., Wang et al. (2023) report a stronger positive geopotential height anomaly over the central North Pacific in January than in November which likely favours wave flux into the stratosphere.
- See new lines L378-381 and 383-384, where we mention the seasonality of the teleconnections.

l. 297: Though I agree with this conclusion, it would be good to emphasize that by day 10 the perturbations in the midlatitudes cover already all longitudes.
- See new line L330

l. 310: This statement is related to my previous remark: Have you investigated for the gradient of vorticity when the forecast error variance reaches a fraction of 0.5, 0.7, and 0.9? Such information or at least some discussion would be highly interesting.
-Please see response above regarding the Rossby Wave Source.

l. 313: This sentence can be removed as it is the same as in line 310.
- agreed. The sentence has been remove

l. 323: I assume that "J" is referring to Judt (2020)? Please clarify.
- Yes – added the full reference.

l. 352: Do you mean with "more globally" simply "globally"? If yes, one could use the operational extended-range predictions of ECMWf as comparison.
- We added a few words to clarify this – see new L393-395

Technical:

We tried to cover all these – see a few details below
l. 72: Please remove "the" before "this".

l. 75: SPPT instead of SPTT
x
l. 112 and elsewhere: Please use the date format as provided in the WCD guidelines:
https://www.weather-climate-dynamics.net/submission.html
- When referring to the initial conditions, we now use 01 January and 01 February instead of abbreviations. But we use a shorthand in referring to the reforecasts, as explained in new L122

l. 124: To be consistent throughout the manuscript, please use the "°" symbol when providing coordinate information.

l. 161: "a" instead of "at"

l. 166 and elsewhere: Please make sure to provide units following the guidelines of WCD.

l. 286 and elsewhere: when indicating ranges between two numbers please use en-dash.

l. 338: Insert blank between "Figure" and "7".

Fig. 4: Is it on purpose that the averaging was performed between 15°-32°N?
Yes- as a kind of compromise between the two latitude ranges shown in Fig. 2

References:
Lee, Y.-Y., and Lim, G.-H. (2012), Dependency of the North Pacific winter storm tracks on the zonal distribution of MJO convection, J. Geophys. Res., 117, D14101, doi:10.1029/2011JD016417.

Deng, Y. and T. Jiang, 2011: Intraseasonal Modulation of the North Pacific Storm Track by Tropical Convection in Boreal Winter. J. Clim., 24, 1122-1137.
https://doi.org/10.1175/2010JCLI3676.1

Moore, R. W., O. Martius, and T. Spengler, 2010: The Modulation of the Subtropical and Extratropical Atmosphere in the Pacific Basin in Response to the Madden–Julian Oscillation. Mon. Wea. Rev., 138, 2761–2779, https://doi.org/10.1175/2010MWR3194.1.
(Rossby wave-breaking)

Quinting, J., Grams, C. M., Chang, E. K.-M., Pfahl, S., and Wernli, H.: Warm conveyor belt activity over the Pacific: Modulation by the Madden-Julian Oscillation and impact on tropical-extratropical teleconnections, EGUsphere [preprint], https://doi.org/10.5194/egusphere-2023-783, 2023.

(WCB)

Wang, J., M. J. DeFlorio, B. Guan, and C. M. Castellano, 2023: Seasonality of MJO Impacts on Precipitation Extremes over the Western United States. J. Hydrometeor., 24, 151–166, https://doi.org/10.1175/JHM-D-22-0089.1.

---

## Author Response (AR3)

Response to Reviewer 2 of the manuscript: **Intrinsic Predictability Limits arising from Indian Ocean MJO Heating: Effects on tropical and extratropical teleconnections**

Reviewer 2 asked that we remove the discussion of the role of ENSO from the first paragraph of Section 4 and remove information regarding ENSO from Table 2 and Section 2.2, instead inserting a statement saying that the state of ENSO is not accounted for due to the small sample size.

We have removed the two discussions of the role of ENSO (in Sections 2.2 and 4 ) and modified Table 2 to remove ENSO information. Very brief statements in line with that suggested are added at:
Lines 136-138 (Section 2.2)
Lines 310-312 (Section 4)
We added the second statement in Section 4 (Discucssion) just to remind the reader that we are not considering ENSO